# Measurements of VOCs in ambient air by GC- and Real-Time-Vocus PTR-TOF-MS: calibrations, instrument background corrections, and introducing a PTR Data Toolkit

Andrew R. Jensen[1,2], Abigail R. Koss[3], Ryder B. Hales[2], Joost A. de Gouw[1,2]

[1]Cooperative Institute for Research in Environmental Sciences, Boulder, Colorado 80309, USA
[2]Department of Chemistry, University of Colorado, Boulder, Colorado 80309, USA
[3]Tofwerk USA, Boulder, Colorado 80301, USA

*Correspondence to*: Joost de Gouw (Joost.deGouw@Colorado.edu)

**Abstract.** Volatile organic compound (VOC) emissions and subsequent oxidation contribute to the formation of secondary pollutants and poor air quality in general. As more VOCs at lower mixing ratios have become the target of air quality investigations, their quantification has been aided by technological advancements in proton-transfer reaction time-of-flight mass spectrometry (PTR-TOF-MS). However, such quantification requires appropriate instrument background measurements and calibrations, particularly for VOCs without calibration standards. This study utilized a Vocus-PTR-TOF-MS coupled with a gas chromatograph for real-time and speciated measurements of ambient VOCs in Boulder, Colorado during spring 2021. The aim of these measurements was to understand and characterize instrument response and temporal variability as to inform the quantification of a broader range of detected VOCs.

Fast, frequent calibrations were made every 2 h in addition to daily multipoint calibrations. Sensitivities derived from the fast calibrations were 5±6% (average and one standard deviation) lower than those derived from the multipoint calibrations due to an offset between the calibrations and instrument background measurement. This offset was caused, in part, by incomplete mixing of the standard with diluent. These fast calibrations were used in place of a normalization correction to account for variability in instrument response and accounted for non-constant reactor conditions caused by a gradual obstruction of the sample inlet. One symptom of these non-constant conditions was a trend in fragmentation, although the greatest observed variability was 6% (one relative standard deviation) for isoprene.

A PTR Data Toolkit (PTR-DT) was developed to assess instrument performance and rapidly estimate the sensitivities of VOCs which could not be directly calibrated on the timescale of the fast calibrations using the measured sensitivities of standards, molecular properties, and simple reaction kinetics. Through this toolkit, the standards' sensitivities were recreated within 1±8% of the measured values.

Three clean air sources were compared: a hydrocarbon trap, zero grade air and ultra-high purity nitrogen, and a catalytic zero-air generator. The catalytic zero-air generator yielded the lowest instrument background signals for the majority of ions, followed by the hydrocarbon trap. Depending on the ionization efficiency, product ion fragmentation, ion transmission, and instrument background, standards' limits of detection (5-s measurement integration) derived from the catalytic zero-air

generator and the fast calibration sensitivities ranged from 2 ppbv (methanol) to 1 pptv (decamethylcyclopentasiloxane; D5 siloxane) with most standards having detection limits below 20 pptv. Finally, applications of measurements with low detection limits are considered for a few low-signal species including sub-pptv enhancements of icosanal (and isomers; 1 min average)

in a plume of cooking emissions, and sub-pptv enhancements in dimethyl disulfide in plumes containing other organosulfur compounds. Additionally, chromatograms of hexamethylcyclotrisiloxane, octamethylcyclotetrasiloxane, and decamethylcyclopentasiloxane (D3, D4, and D5 siloxanes, respectively), combined with high sensitivity, suggest that online measurements can reasonably be associated with the individual isomers.

## 1 Introduction

Volatile organic compounds (VOCs) have been a subject of interest due to their contributions to the formation of secondary pollutants such as ozone (Coggon et al., 2021; Derwent et al., 1996) and fine particles (Li et al., 2021; Odum et al., 1997). Primary emissions from both anthropogenic and biogenic sources as well as their oxidation products affect air quality (Gu et al., 2021) and human health (Nault et al., 2021). Dominant sources of anthropogenic VOCs have shifted in response to mitigation efforts as observed with the reduced automobile emissions (Wallington et al., 2022; Warneke et al., 2012) and

subsequent emergence of volatile chemical products as the largest petrochemical emissions (McDonald et al., 2018). As the atmospheric chemistry community continues to investigate VOC emissions and their chemical evolution, technological advancements have allowed the detection and quantification of a broader range of VOCs at lower concentrations.

Proton-transfer reaction mass spectrometry (PTR-MS) has allowed for real-time detection and quantification of VOCs without the need for pre-concentration or separation (Blake et al., 2009; de Gouw and Warneke, 2007; Hansel et al., 1995;

Lindinger et al., 1998; Yuan et al., 2017b). In this technique, VOCs enter a drift-tube ion-molecule reactor (IMR) and they are ionized via proton-transfer from hydronium ions ($H_3O^+$) produced in a separate ion source. A hollow-cathode ion source is common in PTR-MS instruments, but Tofwerk's Vocus uses a conical, low-pressure discharge source (Krechmer et al., 2018). This transfer takes place for VOCs with proton affinities greater than that of water including most unsaturated hydrocarbons and species with heteroatoms. PTR-MS is insensitive to alkanes due to inefficient proton transfer followed by fragmentation

(Gueneron et al., 2015). Previous studies have demonstrated that PTR-MS sensitivities can be derived from first principles using reactor conditions, rate coefficients, fragmentation, and ion transmission (Holzinger et al., 2019).

Instrument advances have pushed the limits of PTR-MS specificity and sensitivity. The transition from quadrupole to high resolution time-of-flight (TOF) mass analyzers has improved speciation by separating several monoisotopic masses at each nominal mass to yield elemental compositions (Graus et al., 2010; Jordan et al., 2009; Yuan et al., 2016). Chromatography

improved detailed speciation with unit-mass resolution (Fall et al., 2001; Williams et al., 2001; Warneke et al., 2003). The combined application of high mass resolution with chromatography provided another degree of separation and more refined identification. Furthermore, developments in the IMR design have increased sensitivities and lowered limits of detection. For example, the PTR3's reactor design allows for higher pressures and longer reaction times to increase product ion formation

(Breitenlechner et al., 2017). Other IMR designs improve transmission from the drift-tube to the next stage of the mass spectrometer by incorporating ion funnels (Barber et al., 2012; Shaffer et al., 1999) and/or radio frequency electric fields, as in the case of Tofwerk's Vocus IMR (Krechmer et al., 2018) which is used in this study and is discussed further in the next paragraph. These advancements, taken together, allow for the simultaneous and more sensitive detection of a much greater number of compounds in air (Riva et al., 2019; Yuan et al., 2017a).

Krechmer et al. (2018) provide a detailed description of the Vocus design. The Vocus employs a focusing IMR consisting of a glass tube with a resistive coating that is mounted within a radio frequency quadrupole. An axial electric field is applied along the glass tube to enhance ion collision energies and reduce the clustering of ions with water molecules. The quadrupole focuses ions to the central axis of the reactor to improve ion transmission out of the IMR. This design improves detection efficiencies, increases sensitivities ~19 times that of a comparable PTR-MS instrument (Krechmer et al., 2018), and reduces limits of detection. Additionally, the relative flow rates of reagent water vapor and ambient air into the IMR yield high water vapor mixing ratios such that ambient humidity has a negligible effect on the ion chemistry (Krechmer et al., 2018). In contrast, ambient relative humidity impacts the distribution of hydronium and its water clusters as well as analytes' sensitivities of other PTR-MS instruments (Warneke et al., 2001; de Gouw et al., 2003b; Vlasenko et al., 2010; Warneke et al., 2011). Given the divergence in design and operation of the Vocus from traditional PTR-MS instruments, a complete understanding and interpretation of its response, particularly over time, and an evaluation of best practices are necessary.

This study details the quantification of VOCs measured in Boulder, Colorado in spring 2021 with a Vocus-2R PTR-TOF-MS, hereafter referred to as the Vocus, and presents an open-source PTR Data Toolkit (PTR-DT). This study aims to address the production of reliable ambient measurements made by PTR-MS, particularly regarding species lacking standards for calibration. The PTR-DT was used to derive instrument characteristics from measured standards and to estimate the sensitivities of additional species on the timescale of frequent calibrations (every 2 h in this study). Additionally, these frequent calibrations were explored as an alternative for the normalization of ambient measurements against the reagent ion signal, including a time period of changing ion chemistry in the IMR. Three sources of clean air for instrument background measurements were compared to identify which source yielded the lowest limits of detection. Finally, some findings are presented to demonstrate the low detection limits.

## 2 Methods

Measurements were made from a third-story window at the Cristol chemistry building at the University of Colorado Boulder (40.0076° N, 105.2709° W) from March 24th to April 21st, 2021. Water vapor ($H_2O$) and methane ($CH_4$) measurements were made by cavity ring-down spectroscopy (Picarro G2401).

## 2.1 Real-time VOC measurements

VOCs were measured by a Vocus-2R PTR-TOF-MS (Tofwerk AG and Aerodyne Research, Inc.) which is described elsewhere
(Krechmer et al., 2018). The ion source was supplied with a 15 cm$^3$ min$^{-1}$ at STP (25 °C and 1 atm; sccm) flow of water vapor. The IMR was operated at 90 °C and 1.5 mbar with 480 V for the axial voltage and 400 V for the quadrupole amplitude voltage at a frequency of 1.3 MHz, for a reduced electric field strength (*E/N*) of 160 Td. Warmer IMR temperatures improve delay times of species with lower volatilities (Mikoviny et al., 2010). The IMR axial voltage and pressure, and thus *E/N*, were chosen to limit the formation of water clusters and promote simple reaction kinetics, at the expense of analyte fragmentation. The
Vocus also employs a radio frequency "big segmented quadrupole" (BSQ; 255 V amplitude) ion guide in order to attenuate the hydronium ion signal and increase the lifespan of the detector, in addition to focusing the ion beam through an intermediate pumping stage. Mass spectra were collected at a 5-s time resolution from 4–398 Th (TOF extraction frequency of 18.18 kHz). For the measurement period, the mass resolution (m/$\Delta$m FWHM) was ~10,000 for $C_8H_{10}H^+$ (mass-to-charge ratio, *m/Q*, 107).

Prior to all measurements, the instrument's signals were optimized. With a constant flow of a standard mixture,
voltages for the ion optics between the IMR and the TOF mass analyzer were coarsely adjusted to improve overall signal. Finer adjustments were made via Tofwerk AG's Thuner software (v1.13.0.0) which programmatically adjusted voltages and analyzed relative sensitivities and mass resolution. A setpoint was chosen which compromised between high mass resolution and sensitivities.

Ambient air was sampled via a 2-m (0.45 cm ID) polytetrafluoroethylene (PTFE) line at 3.3 L min$^{-1}$ at STP (<1 s
residence time). Air was drawn via an external pump connected to the Vocus inlet such that the sample line led directly into the Vocus inlet (sample flow directed toward the IMR) for subsampling and the excess flow was removed toward the external pump via a perpendicular line also attached to the Vocus inlet (Fig. S1a). Measurements were made on a 2-h cycle consisting of (i) three 2-min instrument backgrounds, (ii) one 2-min calibration, (iii) one 20-min chromatography cycle, and (iv) the remainder reserved for on-line measurements (Fig. S2a). More detailed calibrations and instrument backgrounds were
performed daily (Fig. S2d–e). These different measurements are further detailed in the following sections. Analyses including high resolution peak fitting of mass spectra (Cubison and Jimenez, 2015; Timonen et al., 2016) and TOF duty cycle corrections (relative to *m/Q* 59) were performed using Tofware (v3.2.3; Tofwerk AG and Aerodyne Research, Inc.) in the Igor Pro 8 environment (WaveMetrics, OR, USA).

## 2.2 Instrument troubleshooting

The ion source malfunctioned and was unstable from April 9–10 and 20–21. The electrical current supplying the ion source was highly variable on second timescales and demonstrated a step-change toward higher currents. We believe this was indicative of an incomplete ring in the conical discharge ion source. This issue was resolved by turning off the ion source and flow of water vapor for several minutes before returning to normal operation. The cause is unknown, but a water droplet entering the ion source may be one possibility. Since this discharge was responsible for forming primary hydronium ions, the

consequences of this malfunction were different ion chemistry and reduced sensitivities. For demonstrative purposes, these time periods were included in the discussion of the PTR-DT but are excluded in the final quantified dataset.

Sample air entered the IMR via a short poly-ether-ether-ketone (PEEK) capillary (25 mm; 0.25 mm ID) which serves similarly to a critical orifice. After April 2nd, this capillary was gradually obstructed by particulate matter and the inlet flow rate gradually declined from 205±8 sccm (average and standard deviation of thirteen measurements from March 17th to April

2nd) to 51 sccm on April 10th. Prior solutions have involve using a solvent to clean the capillary or blowing it out with nitrogen, but, at the time of writing, Tofwerk AG recommends replacing the capillary. The capillary was removed and replaced on April 10th, which returned the flow rate to 215±8 sccm (eleven measurements from April 10th to 20th). The IMR pressure was maintained at 1.5 mbar for the full measurement period. The obstruction and changing flow rates were coincident with changing ion chemistry due to the reagent water vapor flow (15 sccm) and the transition from 7% to 23% water vapor by

volume in the IMR, as discussed in Section 4.4. To avoid obstructions of the inlet capillary, we recommend adjusting the geometry of the inlet such that the sample and bypass lines are perpendicular to the Vocus inlet (Fig. S1b).

### 2.3 Instrument background measurements

Instrument background measurements were made by overflowing the Vocus inlet. Excess flow was drawn downstream to the external pump (Fig. S1a) and the main sample line upstream of the Vocus inlet was unaffected aside from reduced flow rates

of ambient air (at most, a reduction of ~0.3 L min$^{-1}$ at STP). Fast, frequent instrument background measurements were made for 2 min approximately every 30 min using a hydrocarbon (HC) trap (VICI Metronics). Additional daily measurements were made with zero grade air initially and later ultrahigh purity (UHP) nitrogen (Airgas) beginning April 1st. Starting April 11th, daily measurements were also made using a newly acquired catalyst-based clean air system (Tofwerk AG) operated at 350 ℃. The zero grade air and UHP nitrogen, collectively referred to as "zero cylinders," as well as the catalyst flushed the Vocus

inlet for 5 min before their respective 5 min measurements (Fig. S2d). The inlets for both the HC trap and catalyst drew from room air.

The instrument background correction applied to each ion was chosen manually according to measurement quality and relative signal. For species with fast responses such as acetone (Fig. S2b), the instrument background signal was derived from the median of the second half of the two-minute measurement. Acrylonitrile demonstrated somewhat longer response

time (Fig. S2b). For species with longer response times, a double exponential function was fit to the data to derive the instrument background. All corrections were applied by linear interpolation (Fig. S2f). Most often, the three clean air sources demonstrated discrepancies in magnitude as discussed in Section 4.5.1.

### 2.4 Calibrations

Both fast, frequent calibrations (e.g., Fig. S2b–d) as well as multipoint calibrations (e.g., Fig. S2e) were performed using

dilutions of a gravimetrically prepared standard mixture of 13 VOCs (Apel-Riemer Environmental, Inc.; Table S1) of nominally 1 ppmv ±5% of each standard (except β-caryophyllene which was 0.1 ppmv ±5%). Fast calibrations were performed

every 2 h by overflowing the Vocus inlet (as described for instrument background measurements) with nominally 8 ppbv of standards. The fast calibrations were buffered with 2 min. of equilibration time prior to the calibration measurement, and 2 min. of purging time with clean air from the HC trap afterwards to remove excess calibrant before recontinuing ambient measurements. Multipoint calibrations from 4–9 ppbv were performed every other day and consisted of five dilutions of the same standard. This range of concentrations was limited by the possible dilution flow rates. Multipoint calibrations of this standard mixture as well as two additional standard mixtures (Apel-Riemer Environmental, Inc.; Table S2; nominally 1 ppmv ±5% of each standard) were also performed before and after the field measurements. Example calibration curves for a multipoint calibration and fast calibration of 1,2,4-trimethylbenzene are shown in Fig. S3.

Following instrument background corrections, the fast calibrations were applied by linear interpolation to the ambient measurements due to higher temporal resolution. Methanol was strongly attenuated by the BSQ, resulting in very low sensitivity. To enhance its sensitivity, the signals of protonated methanol and its cluster with water were summed. A comparison of the fast and multipoint calibrations is discussed in Section 4.2. The measured sensitivities for both fast and multipoint calibrations are available in the data repository (see Data Availability).

**2.5 Gas chromatography**

An Aerodyne Research Inc. gas chromatograph (GC) was interfaced with the Vocus for VOC speciation and determination of fragmentation. Claflin et al. (2021) described a prototype GC design and Vermeuel et al. (2023) described a similar model to that of this study. The GC system used here utilizes a dual-stage thermal desorption preconcentrator (TDPC) consisting of a multi-bed adsorbent sample trap (C3-BAXX-5070; Markes International) followed by a multi-bed adsorbent focus trap (U-T15ATA; Markes International) to improve chromatographic resolution. Separation was achieved with a flow (3 sccm) of UHP helium (Airgas) carrier gas on a Restek MXT-624 column (30 m, 0.25 mm ID, 1.4 μm film thickness), which resolves non-polar to mid-polarity compounds. An oxidant trap was not used in this study.

Figure S4 summarizes the temperature profiles of the GC cycles (20 min) which were performed once every two hours. The cycle begins with a heated backflush of the column. Samples were collected via a 2 m PTFE sample line during real-time Vocus measurements immediately before the start of each cycle. Ambient air was collected onto the sample trap for 10 min at a flow rate of 100 sccm for sample volume of 1 L at STP. Samples were purged for 2 min with clean air from the zero cylinders to reduce the amount of trapped water. Samples were then transferred via carrier to the focus trap then to the column by flash heating the traps in sequence. Mass spectra were recorded at a frequency of 5 Hz during a 10 min chromatogram. To meet the minimum inlet flow rate of the Vocus, the GC effluent was diluted into 250 sccm of air filtered by the HC trap before being directed to the Vocus sample inlet.

Calibrations and instrument backgrounds were performed similarly to on-line measurements. The zero cylinders were used for dilutions and instrument background measurements. The same standard mixture of 13 VOCs was calibrated every other day (Table S1) at nominally 4 ppbv and instrument backgrounds were measured daily. Multipoint calibrations from 0–

12.8 ppbv were performed for the same standards and additional standard mixtures (Table S2) before and after the measurement period and consisted of five dilutions plus an instrument background, each measured in triplicate.

Chromatographic peaks were analyzed using the TERN software package (v2.2.19; Aerodyne Research, Inc.) in the Igor Pro 8 environment (Lerner et al., 2017). Following high resolution peak fitting in Tofware, the data were imported into TERN where peak areas were determined by fitting chromatographic peaks (Isaacman-VanWertz et al., 2017).

## 3 Determination of sensitivities: the PTR-DT

Direct calibration of a standard is preferred, but standards are not available for all analytes one may wish to quantify. In the absence of standards for specific analytes, available standards for other analytes may be used to characterize instrument response and estimate the sensitivities of other analytes of interest. The role of the PTR-DT is to use measurements of standards to estimate the sensitivities of other species on the same timescales of the field calibrations. The PTR-DT is made available as an Igor procedure file (.ipf) and is accompanied by an Igor formatted notebook file (.ifn) which provides more detailed instructions. Currently, version 1.1 is known to function in Igor Pro 8 and 9. This toolkit can be modified to fit the needs of other use cases or be expanded to address additional aspects of PTR-MS quantification. The performance and limitations of the toolkit are discussed in Section 4.1.

Sensitivity estimates in this toolkit rely on simple reaction kinetics. The concentration of product ions, $[RH^+]$, formed in the IMR given a concentration of a VOC, $[R]$, is given in Eq. (1):

$$[RH^+] = [H_3O^+]_0 \times \left(1 - e^{-k_{PTR}[R]t}\right) \approx [H_3O^+]_0 \times [R] \times k_{PTR} \times t, \tag{1}$$

where $[H_3O^+]_0$ is the initial concentration of hydronium in the IMR, $k_{PTR}$ is the PTR rate coefficient, and $t$ is the residence time of hydronium ions in the IMR. For a short reaction time and negligible depletion of reagent ions, product ion formation is approximately linear with the concentration of the VOC. The sensitivity of $R$, $S$ (in counts per second per ppbv; cps ppbv$^{-1}$), is defined as the number of ions produced for a unit trace gas mixing ratio, as shown in Eq. (2):

$$S = \frac{[RH^+]}{[R]} \approx [H_3O^+]_0 \times k_{PTR} \times t. \tag{2}$$

Holzinger et al. (2019) have demonstrated that PTR-MS instruments follow these simple reaction kinetics under typical operational conditions and after accounting for other influential factors, allowing for the quantification of uncalibrated organics. The number of measured product ions ($[RH^+]_{meas}$) is attenuated by the fragmentation of the quantitative ion as well as the ion optics of the instrument, duty cycle of the mass analyzer, and tuning/aging of the detector (Müller et al., 2014). Equation (3) incorporates the fraction of signal attributed to the quantitative ion, $f$, and an $m/Q$-dependent transmission function, $T(m/Q)$, yielding the instrument sensitivity, $S_{inst}$:

$$S_{inst} = \frac{[RH^+]_{meas}}{[R]} \approx [H_3O^+]_0 \times k_{PTR} \times t \times f \times T(m/Q). \tag{3}$$

Assuming no additional, outside factors, e.g., passivation effects and spectral interference, then $S_{inst}$ is expected to equal the measured sensitivity, $S_{meas}$. Here, spectral interferences refer to contributions to an analyte's quantitative ion from the fragmentation and/or adducts of other ions (for example, ethylbenzene commonly fragments to form $C_6H_7^+$, contributing additional signal to that of protonated benzene). Equation (3) is a simplification since atmospheric measurements are complex and interferences are common. The PTR-DT does not account for spectral interference as discussed in Section 4.1.6.

Each step of the toolkit is broadly described in Sections 3.1–3.5, including the consideration and choices used in the quantification of the Boulder 2021 measurements. Screenshots of the interfaces for Steps B–D are shown in Fig. S5. Briefly, the toolkit is divided into five steps:

A. Initialization of the toolkit and data entry,
B. Characterization of instrument sensitivity changes between field and laboratory calibrations for the purpose of calibrating additional standards solely in the laboratory,
C. Characterization of instrument sensitivity as a function of $k_{PTR}$,
D. Characterization of the instrument's $T(m/Q)$, and
E. Estimation of sensitivities using the characterization from previous steps for compounds which were not directly calibrated.

**3.1 Step A: initialization and data entry**

Step A involves data entry for calculations in later steps. Some key parameters, namely $f$ and $k_{PTR}$ are discussed here. Experimental values of $k_{PTR}$ are typically scarce, particularly for exact instrument operating conditions of a given set of measurements (for example, *E/N* of 160 Td). Instead, they can be estimated given molecular properties. In addition to *E/N*, $k_{PTR}$ depends on molecular polarizability and permanent dipole moments (Langevin, 1905; Chesnavich et al., 1980; Su and Chesnavich, 1982; Su, 1994) which are available in the literature for more species (Cappellin et al., 2010, 2012; Haynes, 2014; Langford et al., 2013; Zhao and Zhang, 2004). Moreover, Sekimoto et al. (2017) have parameterized these molecular properties for a wide range of functional groups, elemental compositions, and mass. In this study, all values of $k_{PTR}$ were calculated based on the reactor conditions as well as molecular polarizability and permanent dipole moments from the literature, if available, or otherwise estimated based on Sekimoto et al. 's (2017) parameterizations.

**3.1.1 Determining fragmentation**

Gas chromatography was used to determine $f$ for resolved standards and speciated compounds by taking the ratio of the quantitative ion peak area ($A_Q$) to the sum of the areas of the quantitative ion, clusters with water molecules, and fragments ($A_i$), as shown in Eq. (4):

$$f = \frac{A_Q}{\sum A_i / I_i}. \tag{4}$$

To account for minor isotopologues, the areas in the sum were also scaled according to their isotopic abundance ($I_i$). The minor isotopologues are assumed to undergo proton-transfer with similar values of $f$ and $k_{PTR}$. Signals attributed to charge transfer products were not included as they represent a different ionization pathway (e.g., $O_2^+$ or $NO^+$). The inclusion of water clusters was inconsequential in this study due to the high *E/N* such that water cluster contributions were less than 3% across all standards. Figure S6 shows example chromatograms of the standards in Tables S1 and S2 as well as the fragments used to calculate their quantitative ion fractions. Pagonis et al.'s (2019) PTR Library tabulates PTR-MS observations including fragmentation information with various instrument parameters as reported in the literature.

Values of $f$ were determined to account for all ions produced by PTR in the IMR. While standards are directly calibrated, their fragmentation rates are necessary to characterize the simple reaction kinetics in the later stages of the PTR-DT. To estimate the sensitivities of analytes which cannot be directly calibrated, the estimated sensitivities must also be corrected for fragmentation. In the absence of direct calibration, sensitivity uncertainties resulting from ignoring fragmentation may range from negligible (acetonitrile) to a factor of 4 (decamethylcyclopentasiloxane), although fragmentation was enhanced by the high *E/N* in the study. In the absence of standards, if a GC is available, $f$ can be quantified for identified chromatographic peaks 6. Without a GC to quantify fragmentation, an estimate from the literature or an informed assumption (e.g., from an analogous compound or from a database) may be preferred. In either case, untargeted analyses are still possible and reasonable accounting for fragmentation will improve the accuracy of quantified mixing ratios.

### 3.1.2 Considerations regarding fragmentation

This fraction is expected to be constant provided *E/N* remains constant, so it is treated as such in the toolkit. This assumption is discussed in Section 4.1.1. In this study, an *E/N* of 160 Td was used to limit the frequency of proton-transfer with hydronium-water clusters and ensure simple reaction kinetics. This *E/N* was greater than the more typical *E/N* of 120 Td, which itself was reported to cause significant fragmentation and limit the subset of observable VOCs (Riva et al., 2019). In Tables S1 and S2, the values of $f$ represent the average fractions from all field and laboratory calibration chromatograms, respectively. Greater uncertainties were observed for low signal-to-noise, poorly resolved, and highly fragmented species. Alternatively, $f$ may be left as unity while the sum of ambient fragment signals is quantified if spectral interference is negligible.

Some standards, e.g., isoprene and cyclohexene, and their fragments did not have the same transmission efficiency due to a sharp mass cutoff below *m/Q* 60 in the BSQ following the IMR. Values of $f$ should reflect the product ion distribution in the IMR rather than the measured distribution. Without accounting for transmission efficiency for these fragments, the sum of all ions produced by a standard's ionization would be underestimated and $f$ as well as the calculated sensitivity would be overestimated. Such standards were temporarily excluded in steps C and D. However, the transmission function was determined using the remaining standards then used to scale the peak areas in Eq. (4) to determine $f$ for those excluded standards. After including these standards in steps C and D, the transmission function and values of $f$ were refined. This process can be done iteratively but was only done once here as no significant change in transmission was observed. Other

standards also had fragments with different transmission efficiencies, e.g., acetone, but were not excluded due to their negligible contributions to the total signal.

Thermal decomposition during injection and chromatography is a possibility, particularly following flash heating to 300 ℃ (Fig. S4), although such products were not observed during this study. Decomposition products would be expected to arrive at different retention times than the parent and not affect the observed fragmentation rate in the Vocus itself.
Additionally, losses of analyte in the GC system, related to decomposition or otherwise, would not affect $f$ since all peak areas are relative to parent ion.

### 3.2 Step B: estimating field sensitivities of laboratory standards

In field studies, instrument sensitivity is often monitored using a single calibration standard mixture, while the instrument is characterized in more detail before and after the study using additional mixtures. The purpose of step B is to relate the field
sensitivities to those calibrated in the laboratory. In doing so, these laboratory standards provide a greater subset of directly calibrated VOCs to quantify the ambient time series as well as to constrain the later characterization steps. This step is optional if such additional measurements are not available.

The field sensitivities were regressed against the laboratory sensitivities of the same standards as to minimize orthogonal distance. Figure 1a shows this regression for the first field calibration and the pre-field laboratory calibration (Fig.
S7a shows a similar regression using post-field laboratory calibrations). Fits using pre- and post-field calibrations yielded similar results. The post-field calibrations had lower sensitivities, yielding a higher slope. Methanol was excluded from all regressions due to near-zero and sometimes negative laboratory sensitivities; $m/Q$ 33 is too far below the mass cutoff of the BSQ. The toolkit allows for a customizable inclusion table where individual standards may be excluded for individual calibrations as necessary. Fit parameters were retrieved on the same 2 h timescale as the field calibrations and temporal trends
are discussed in Section 4.1.2. These parameters were then used to convert the measured laboratory sensitivities of the standards in Table S2 to estimated field sensitivities. Uncertainties in these laboratory sensitivities, here the standard deviation of replicate measurements, were propagated with the uncertainties of the regressions. This process was repeated with both the pre- and post-field calibrations, then the two sets of estimated field sensitivities were averaged and applied by linear interpolation.

### 3.3 Step C: relating sensitivities to simple reaction kinetics

The simple reaction kinetics in Eq. (2) show that PTR-MS sensitivities are directly proportional to $k_{PTR}$ provided other influencing factors are negligible or corrected. Step C investigates this linear relationship empirically, using the field standards' measured sensitivities as well as the lab standards' field-estimated sensitivities. These sensitivities are corrected for fragmentation to account for all ions that have undergone proton transfer. The toolkit includes a low- and high-$m/Q$ cutoff to
310 account for reduced transmission efficiencies. Standards influenced by additional factors, e.g., passivation and spectral

interference, must be manually excluded. The slope of the resulting linear fit is the product $[H_3O^+]_0 \times t$ such that neither quantity is necessary to complete these calculations.

Figures 1b and S7b show orthogonal distance regressions for the first field calibration using field-estimated sensitivities of laboratory standards informed from the pre-field and post-field laboratory calibrations, respectively, each yielding similar results. From Eqn. (3), the y-intercept is expected to be zero. Here, the intercept is arbitrarily not forced through the origin. These regressions provide an expected sensitivity ($S_{exp}(k_{PTR})$) which accounts for all product ions prior to fragmentation, attenuation, or other factors. In this study, the acceptable $m/Q$ range was 58–150 Th. Other standards were excluded due to unconstrained fragmentation (phenol and β-caryophyllene), poor passivation of the calibration line (d-limonene), and spectral interference (first acetone in Table S2, both benzenes in Table S2, toluene).

## 3.4 Step D: determining the transmission function

The ratio of a standard's $S_{meas}$ and $S_{exp}(k_{PTR})$ can be used to estimate that ion's relative transmission ($T_R$; Eq. 5). These ion transmissions are then used in step D to retrieve the instrument's transmission function, $T(m/Q)$, which is fit by the product of two sigmoid functions (Eq. 6):

$$T_R = \frac{S_{meas}}{S_{exp}(k_{PTR})}, \tag{5}$$

$$T(m/Q) = \left[1 + exp\left(\frac{M_L - m/Q}{w_L}\right)\right]^{-1} \times \left[1 + exp\left(\frac{M_H - m/Q}{w_H}\right)\right]^{-1}, \tag{6}$$

where $M_L$ is the cutoff $m/Q$ (50% transmission) and $w_L$ is the rate of the transmission change, both in the low $m/Q$ range while $M_H$ and $w_H$ are the analogous fitting parameters in the high $m/Q$ range. The base and maximum parameters of both sigmoidal fits are set to 0 and 1, respectively. This model is similar to that applied to several PTR-MS instruments by Holzinger et al. (2019), except the PTR-DT does not include a term for the TOF duty cycle which was instead corrected in Tofware. The low $m/Q$ range was affected by the BSQ's transmission attenuation. A high $m/Q$ mass discrimination is introduced by the quadrupole ion guides due to slower velocities and non-uniform fields near the entrance and exit of the quadrupoles (Antony Joseph et al., 2018; Dawson, 1975; Fite, 1976; Ehlert, 1970). Additionally, aging or poor tuning of a multichannel plate detector may reduce the relative detection efficiency at higher $m/Q$, resulting in mass discrimination (Müller et al., 2014). Absolute detection efficiencies are negatively correlated with $m/Q$ when not operating the detector in saturation mode (that is, the electron cascade is in saturation regardless of the ion's $m/Q$) (Oberheide et al., 1997). Typically, PTR-TOF-MS users do not operate in saturation mode due to artefacts such as ion feedback (Pan et al., 2010). To account for reduced transmission and detection efficiency in the high $m/Q$ regime, a second, optional sigmoid function is available in the toolkit. Separate, customizable $m/Q$ subranges are used when fitting the two sigmoid functions.

Figures 1c and S7c show example transmission functions derived from the first field calibration using field-estimated sensitivities of laboratory standards informed from the pre-field and post-field laboratory calibrations, respectively, each yielding similar results. Uncertainties in the values of $T_R$ were derived from the propagation of the uncertainties from $S_{meas}$,

which includes propagation through the uncertainties of the regressions in step B for laboratory standards, and the uncertainty of $S_{exp}(k_{PTR})$ from the regressions in step C. Some standards were excluded due to similar reasons listed for step C (Section 3.3) and additional standards, previously excluded due to their *m/Q*, were excluded in this step due to poorly constrained fragmentation (methanol and ethanol) and poor passivation (acrolein). In the high *m/Q* range, relative transmission was ~1 up to *m/Q* 297 (octamethylcyclotetrasiloxane; D4 siloxane), but decamethylcyclopentasiloxane (D5 siloxane; *m/Q* 371) had a reduced relative transmission of ~0.7 (Fig. 1c).

### 3.5 Step E: sensitivity estimations

The main purpose of Step E is to determine the sensitivity for those compounds that lack a calibration standard. The relationship between sensitivity and rate coefficient from step C, the instrument transmission function from step D, and information about the VOC to be calibrated are used in Eq. (7) to calculate the estimated sensitivity ($S_{est}$):

$$S_{est} = S_{exp}(k_{PTR}) \times T(m/Q) \times f. \tag{7}$$

While the transmission function is retrieved for each calibration, the average transmission function can optionally be used instead. The quality of the sensitivity estimation is explored by calculating $S_{est}$ for all standards for comparisons against the measured values. The performance of the PTR-DT for this study is discussed in Section 4.1.

For each VOC to be calibrated, inputs for $k_{PTR}$ and $T(m/Q)$ are necessary while $f$ is ideal if available. If $f$ cannot be determined, the upper bound for that VOC's sensitivity is calculated. For this study, the pre- and post-field laboratory calibrations were used in the PTR-DT separately, using the same settings during each step. The final sensitivities were then averaged before application to the ambient time series, as was described for the laboratory standards. Given sufficient standards to characterize the instrument's response and sufficient knowledge of the VOC to be calibrated, it becomes possible to calibrate any measured VOC.

## 4 Results and discussion

### 4.1 Performance and limitations of the PTR-DT

Figures 2 and 3 summarize the fitting results and performance (i.e., accuracy and precision) of the PTR-DT, respectively. The left column shows the time dependence of the scaling between the in-field calibrations versus those performed before the study. The middle column shows the time series of the scaling between the in-field calibrations and the proton-transfer rate coefficients. And the right column illustrates the time series of the parameters that define the mass transmission in the low *m/Q* regime. Similar plots are generated during each step of the toolkit for quick assessment of performance and allow for optimization prior to the next step.

The time series of fitting parameters in Fig. 2 provide a sense of instrument performance and temporal stability. Between March 24 and April 7, the fit parameters changed gradually, but the quality of the fits between field and laboratory

sensitivities remained good ($R^2 > 0.95$) and the PTR-DT accounted for the drifts in instrument sensitivity. The periods where the ion source malfunctioned (April 9–10 and 20–21) demonstrated significantly different fitting parameters and the step change informed of a sudden, undesirable change in instrument operation. After disassembling the inlet to replace the capillary and resetting the ion source voltages, the ion chemistry returned to a similar state as evidenced by strong correlations, in particular between field and laboratory sensitivities. The instrument did not return to the same overall sensitivity, but all compounds were affected similarly. The instrument can recover from malfunctions and maintenance, returning to a similar state of response, but recalibration is critical.

The relative residual histograms for each step in Fig. 3 show that the regressions and estimated sensitivities generally recreated the measured or derived counterparts, although there are outliers as detailed later. The residuals for steps B–D compare the measured or derived values against those estimated from the regressions. The residuals for step E compare the measured or field-estimated sensitivities against those calculated from the input parameters and the regressions throughout the PTR-DT.

### 4.1.1 Fragmentation variability

Many uncertainties and limitations in this toolkit, and more broadly PTR-MS quantification, stem from uncertainties in the key quantification parameters in step A. This toolkit assumes $k_{PTR}$ and $f$ are constant, which was not the case for this study with non-constant reactor conditions which are discussed further in Section 4.4. Independent of variable reactor conditions, accurate quantification requires accurate determination of $k_{PTR}$ and $f$.

Figure 4 shows that the quantitative ion fractions slowly drifted during the period where the Vocus inlet flow was gradually reduced and the ion chemistry varied but was otherwise stable prior to the inlet obstruction and after the replacement of the PEEK inlet capillary. Less fragmentation was observed during the latter half of the measurement period, possibly due to the exact positioning of the PEEK capillary and corresponding introduction of sample air to the IMR. The greatest variability in $f$ was observed for transmission-corrected isoprene with a quantitative ion fraction of 0.39±0.02 (6% relative standard deviation) averaged across all GC field calibrations. Additional variability in isoprene's quantitative ion fraction was introduced by the transmission correction. Several fragments fell in the low *m/Q* regime where the sharp transmission drop-off occurs. This sigmoid was constrained by relatively few standard compounds (Fig. 1c) and small uncertainties in this fit lead to large transmission uncertainties. Regardless, α-pinene had the next largest relative standard deviation of 5%.

While variable fragmentation may not always be negligible, these observations suggest that fragmentation can be treated as approximately constant under reasonably constant reactor conditions. With this assumption, fragmentation may be probed less frequently. However, for quality assurance, it would be prudent to reevaluate fragmentation rates with any significant changes such as replacing the inlet capillary, tuning the instrument, or moving the instrument. At minimum, it is recommended to quantify these fragmentation rates at the beginning and end of field measurements, if possible.

### 4.1.2 Trends in instrument response

The regressions from step B characterize the general changes in instrument sensitivity over the course of the field measurements. The slope was initially ~1 due to the short time period between the laboratory calibrations and the early field calibrations. The slope then increased due to generally increasing sensitivities in the field, driven by changing ion chemistry as discussed in Section 4.4, then the slope and linearity ($R^2$) declined when the ion source malfunctioned. Outside these periods, the correlations between pre-field and field sensitivities were generally strong ($R^2 > 0.95$; Fig. 2) which indicate that all calibrants nominally responded to changes in instrument conditions and hydronium ion concentrations similarly. However, linearity demonstrated a gradually decreasing trend as the inlet was obstructed due to changing ion chemistry and water cluster distributions, as discussed in greater detail in Section 4.4.

Benzene and toluene, for example, react slowly or not at all with hydronium-water clusters due to insufficient proton affinities (Warneke et al., 2001), and experienced gradually increasing positive residuals between the measured and predicted sensitivities during this period (meaning that predicted sensitivities were higher). Simultaneously, acetone, among others, experienced gradually decreasing negative residuals. The sensitivities of benzene remained nominally constant while those of acetone gradually increased (Fig. S8). Removal of benzene and toluene improved linearities and reduced residuals of all other standards. That is, benzene and toluene had a different response to the changing ion chemistry relative to acetone and other standards, likely due to, in part, the relative reaction rates with hydronium and water clusters which are changing during this time. The changing ion chemistry is discussed in greater detail in Section 4.4. To account for this difference in response, other studies have applied empirical, VOC-specific corrective factors when normalizing product-ion signals against hydronium and hydronium-water cluster signals (Warneke et al., 2003; de Gouw and Warneke, 2007). When the PEEK inlet tubing was replaced, the ion chemistry returned to a similar state as evidenced by the improved linearity.

Fitting a normal distribution to the residual histogram indicated that the estimated sensitivities recreated the measured sensitivities within -2±3% (average and standard deviation of the fitted distribution; Fig. 3). β-caryophyllene was an outlier with relative residuals ranging from 8% to -105% which were attributed to its low measured sensitivity and the relatively uncertain fit of the y-intercept. Additionally, the slow passivation of β-caryophyllene may contribute to these residuals. The tailing toward positive residuals was primarily caused by the changing ion chemistry affecting benzene and toluene. Additionally, α-pinene had a consistent residual of ~8%.

### 4.1.3 Simple reaction kinetics

The step C fitting parameters characterize the relationship between measured and estimated field sensitivities with $k_{PTR}$. This step requires a set of standards with a sufficient range of $k_{PTR}$ values and functional groups in order to make meaningful approximations of sensitivities for other species. This study included values of $k_{PTR}$ ranging from 1.68–3.82 $\times$ 10$^{-9}$ cm$^3$ molec$^{-1}$ s$^{-1}$ between the field and laboratory standards. The parameters demonstrated a similar temporal behavior to those in step B

(Fig. 2) since both characterize the temporal variability of field sensitivities which have linear relationships with the laboratory sensitivities and $k_{PTR}$.

Step C's regressions demonstrated good linearity ($R^2 > 0.78$; excluding periods where the source malfunctioned; Fig. 2) and recreated similar sensitivities within 3±8% (Fig. 3). The skewed, positive residuals were driven by α-pinene and benzene with residuals ranging from 27–47% and 20–40%, respectively, during the former portion of the measurements as the inlet capillary became obstructed. Benzene's high residuals were attributed to the changing ion chemistry as described in the previous paragraph. With a few exceptions, the majority of residuals were within 20%.

### 4.1.4 Transmission

The transmission function's mass cutoff and rate of change for this cutoff are characterized by step D's fitting parameters. Excluding the periods with the malfunctioning source, $T(m/Q)$ was generally stable during these field measurements with some long-term trends (Fig. 2) where such long term trends have been observed elsewhere (Taipale et al., 2008). Although not applied here, the PTR-DT allows for an average $T(m/Q)$ to be applied for shorter field measurements. On average, the PTR-DT estimated relative transmission efficiencies within 2±7% (Fig. 3). The largest residuals belonged to D5 siloxane, which accounted for nearly all residuals greater than 40%, due to the unconstrained transmission and detection efficiency in the high *m/Q* regime. Aside from D5 siloxane, α-pinene and benzene again account for the majority of the residuals beyond ±20%.

Characterization of different mass regimes requires an adequate subset of standards which span these regimes. At high *m/Q*, only D4 and D5 siloxanes were included in our standards with relative transmissions of ~1 and ~0.7, respectively (Fig. 1c). Insufficient information was available to constrain a second sigmoidal fit in this high *m/Q* regime. Due to this limitation, transmission and detection efficiency in this regime was assumed to be ~1. This assumption is reasonable at least to *m/Q* 297 (D4 siloxane), but the estimated transmissions and sensitivities beyond this *m/Q* were overestimated. Relatively few species were quantified in this *m/Q* range and uncertainties in fragmentation were more significant than uncertainties in transmission. Regarding species that define the cutoff mass and rate of change, determination of $T(m/Q)$ was dependent on step C such that variability in the regression of measured sensitivities against $k_{PTR}$ manifested as variability in $T(m/Q)$.

### 4.1.5 Reconstructed sensitivities

The residual plot for step E shows the PTR-DT, on average, estimates similar sensitivities as measured for the standards within 1±8% (Fig. 3). All standards in Tables S1 and S2 with fragmentation information were included (excluded: methanol, ethanol, phenol, t-amyl ethyl ether, β-caryophyllene). The standard deviation of these residuals is biased low as the measurements were used to construct the regressions which then determined the estimated sensitivities. However, this histogram includes species with known spectral interference (toluene, acetone in the same standard mixture as t-amyl ethyl ether, and both benzenes in Table S2) which have high residuals. In order of magnitude, the greatest residuals belonged to d-limonene, D5 siloxane,

acrolein, α-pinene and benzene for reasons described above. Aside from these outliers, the majority of standards' residuals fell within ±20%.

### 4.1.6 Overall evaluation

The PTR-DT is limited in that it does not account for back reactions as is necessary for formaldehyde and similar compounds with proton affinities only slightly greater than that of water (Vlasenko et al., 2010; Warneke et al., 2011). This back reaction requires further investigations with regards to the Vocus due to the abundance of water vapor in the IMR. Such species' calculated sensitivities are overestimated, and corrections must be made separately.

Additionally, the PTR-DT does not account for spectral interference. That is, the fragmentation or adduct formation of other species increase the measured signals of a target analyte. Sensitivities from the PTR-DT, which correspond to the target analyte alone, will yield overestimated concentrations. Values of $k_{PTR}$ used in the PTR-DT will also only correspond to the target analyte and have no relation to interfering fragment ions. However, these limitations are not unique to the PTR-DT and also apply to the use of standards to measure sensitivities. To account for these interferences, analyte- and interference-specific corrections could possibly be applied to the estimated sensitivities, but these interferences may be on shorter timescales than routine calibrations. Instead, corrections informed by GC may be applied to the real-time signals as demonstrated by Vermeuel et al. (2023) for aldehyde fragmentation contributions to isoprene's quantitative ion. Briefly, they used GC to characterize the relative abundance of $C_5H_9^+$ (the quantitative ion used for isoprene) compared to the parent ions for *n*-aldehydes. Then, they scaled the real-time signal for those aldehydes by that relative abundance and subtracted those contributions from the real-time signal for $C_5H_9^+$. The remaining signal uniquely corresponded to isoprene and was calibrated using the isoprene sensitivity.

A few key points are useful in the interpretation of these residual histograms. While standards with unknown fragmentation rates were excluded in these histograms, fragmentation may be poorly constrained for some species. This was particularly true in the low *m/Q* regime where parent ions and fragments have different transmission efficiencies. Without the application of a transmission correction, fragmentation would be underestimated which would result in positive residuals. Overestimation and underestimation of fragmentation resulted in negative and positive residuals, respectively, but underestimation may be more likely to occur. Moreover, the treatment of $k_{PTR}$ and $f$ as constants introduced trends in the residuals of susceptible species and reduced the accuracy and precision of estimated sensitivities. Spectral interferences cause an overestimation of measured sensitivities, yielding negative residuals. Passivation of the calibration line was dependent on the line material and length as well as the compound's volatility and Henry's Law constant (Deming et al., 2019; Liu et al., 2019; Pagonis et al., 2017). Insufficient passivation time reduces the measured sensitivities and yields positive residuals. Passivation effects were also strong for acrolein and are expected for other compounds with strong interactions with surfaces (e.g., aldehydes and amines). Unlike standard calibrations, ambient measurements involve a mixture of isomers which may have different values of $k_{PTR}$ and $f$. The PTR-DT can estimate sensitivities for speciated isomers provided the necessary parameters. Otherwise, similar elemental compositions tend to have similar dipole moments as well as polarizabilities which

correlate well with mass (Sekimoto et al., 2017), thus yielding similar rate constants. However, $f$ can vary significantly across isomers, e.g. the loss of a water molecule is common for $n$-aldehydes and less so for comparable $n$-ketones (Buhr et al., 2002; Pagonis et al., 2019).

In this study, the histograms were all skewed toward positive residuals as the majority of these factors contributed to positive residuals. The strongest contributor to negative residuals, spectral interference, was limited by the removal of affected standards. Additionally, similar standards demonstrated high residuals across the different steps of the PTR-DT (e.g. α-pinene and benzene) as each step depended on the previous. Despite the assumptions and limitations, the PTR-DT performed well and was able to estimate sensitivities accurately with reasonable precision.

## 4.2 Calibration accuracy

Quantification in this study has relied on the 2 h resolution calibrations to address short-term variability in instrument sensitivity. These fast calibrations used two measurements: one moderate dilution of a standard mixture, and one instrument background measurement. The resulting sensitivities depended heavily on these two reference points, so accuracy is addressed here. Figure 5 compares the sensitivities derived from multipoint and adjacent fast (±4 h; not averaged) calibrations of the standards in Table S1. Data are presented as the departure from zero, where 0 means the two calibrations agree and -10% means the fast calibration sensitivity is 10% lower. Across these standards, the average residual with one standard deviation was -5±6%. Except for methanol, all compounds demonstrated systematically lower fast calibration sensitivity relative to the multipoint calibration sensitivities. The HC trap was used for both the instrument background and calibration measurements such that excess signal from the HC trap would manifest as a constant offset in all signals (discounting <1% additional dilution via addition of the standard) and not affect the slope, i.e., the sensitivity. Moreover, the measured instrument background and calibration signals agreed between the two types of calibration.

### 4.2.1 Discrepancy between fast and multipoint calibrations

Instead, an offset was found between the instrument background and calibration measurements such that the regression of the calibration measurements (excluding the instrument background measurement) had a negative y-intercept of -1700±400 cps (Fig. S3; error reflects uncertainty in the linear fit). The multipoint calibration sensitivities used in Fig. 5 exclude the instrument background measurement in the regressions due to this offset. The fast calibrations relied on the instrument background measurement and had a y-intercept of >0 cps. Inclusion of the instrument background measurements in the regressions reduced the residuals between the two types of calibrations and brought the y-intercept closer to 0, but also reduced the quality of the regression. In short, the residuals in Fig. 5 are due to the method of deriving the sensitivity.

The offset was standard-dependent and does not appear to be caused by poorly constrained flow rates (all mass flow controller flow rates were verified). Incomplete mixing of the standard and diluent was one possible contributor. Figure S9 shows a minor correlation between standards' average residuals and their diffusion coefficients in air (Yaws, 2008), although there are likely other factors as well. Lower volatility standards demonstrated greater residuals, but no clear dependence on

standards' saturation vapor concentration was observed and inlet passivation does not seem to be the primary cause for the residuals.

The average residual of -5±6% was acceptable given the >±10% uncertainty typically reported for PTR-MS measurements as well as the uncertainty caused by possibly incomplete mixing or other factors. A possible solution may involve changing the mixing geometry for the dilution of standard mixtures. Alternatively, an additional fast calibration could be made such that the sensitivities are derived from two standard dilutions rather than a single dilution and an instrument background measurement.

The frequency of fast calibrations in this study were sufficient to capture variability in instrument response. Instrument response stability benefited from being at a ground site in a controlled setting. Mobile, aircraft, or eddy covariance measurements as well as any extreme conditions may require more frequent fast calibrations or another method of correcting for variability in instrument response.

## 4.3 Normalization

In PTR-MS, ambient signals and sensitivities are commonly normalized to the reagent ion signals to correct for hydronium ion production variability and ambient relative humidity. Proton-transfer can occur between a VOC and hydronium as well as its clusters with water molecules ($H_3O+(H_2O)_n^+$; n = 0, 1, 2, …) provided sufficient VOC proton affinities. Normalization factors often include a compound-dependent linear combination of the dominant water cluster signals, typically n=0 and n=1 (de Gouw et al., 2003a; de Gouw and Warneke, 2007), due to the unequal protonation via the water clusters. Normalized sensitivities should be approximately constant provided that the variability in sensitivities were driven by the relative abundance of primary ions as opposed to different measurement conditions.

### 4.3.1 Issues with normalizing Vocus signals

For this dataset, signals and sensitivities were not normalized to the reagent ion signal. The Vocus IMR contains a high mixing ratio of water such that the hydronium ion signal and sensitivities do not depend on sample relative humidity (Krechmer et al., 2018) which was observed in this study (Fig. S10a–b). The attenuation caused by the BSQ complicates the retrieval of the true hydronium ion abundance due to the lack of standards in this very low $m/Q$ regime to determine transmission efficiency. Normalization against $m/Q$ 19 (unit mass of $H_3O^+$), $m/Q$ 37 ($H_5O_2^+$, the hydronium-water cluster), or some combination thereof did not account for the time dependence of the benzene and acetone sensitivities simultaneously. Here, an example normalization against $m/Q$ 19 exacerbated rather than mitigated variabilities in sensitivities for acetone and benzene (Fig. S8). The unit mass for hydronium was used due to difficulties with accurate mass calibration and peak integration at this low $m/Q$. VOC sensitivities in this study were not driven mostly by primary ion concentrations, but rather were driven by non-constant IMR conditions as discussed in Section 4.4.

### 4.3.2 Alternatives to normalization

Rather than normalize ambient signals, this study instead relied on fast, frequent calibrations to account for variability in primary ion concentration and other instrument variabilities. In doing so, these calibrations are assumed to be on shorter timescales than such changes.

One possible modification to this method of fast calibrations may involve a constant introduction of an internal standard which is otherwise not present in the sampled air, e.g. deuterated acetone or benzene. This method, in a sense, brings

fast calibrations to the extreme of "calibrating" with each mass spectrum. While this method was not used in this study, we make a few suggestions following our observations. Further investigation and validation is required. Multiple standards may be necessary to account for different dependencies on hydronium and its water clusters. Additionally, care should be taken in assessing fragmentation of such standards and how they may interfere with analytes. Finally, we strongly encourage the use of internal standards *in addition* to traditional calibrations with more standards.

### 4.4 Obstruction of the inlet capillary

Due to the use of a PEEK capillary as a critical orifice in the Vocus inlet, obstruction and declining sample flow are common problems frequently noted in the Vocus community. Specifically, capillary obstruction commonly occurs when sampling air polluted with particulate matter including chamber studies with high secondary organic aerosol yields. First and foremost, this issue can be mitigated by the introduction of a tee to the inlet such that the sample and bypass lines connect in sequence and

575 are perpendicular to the inlet (Fig. S1b). If an inlet obstruction is not addressed, VOC sensitivities may vary with different responses across different species. As the inlet was obstructed in this study, the sample inlet flow transitioned from 205±8 to 51 sccm, as described in Section 2.2. This gradual obstruction coincided with a ~21% enhancement in acetone's sensitivity while benzene's sensitivity saw a ~9% reduction. Most other standards and ambient signals saw similar trends to varying degrees of severity. Such significant and variable changes in instrument response serve as indicators that something is wrong

and must be addressed. Accurate quantification depends on consistent ion chemistry in the IMR and variable reactor conditions should be avoided where possible. Here, we document some observations concurrent with variable IMR conditions, and discuss some of the effects contributing to these observations and variable sensitivities.

### 4.4.1 Observations suggesting variable IMR conditions

Trends in hydronium ion signal (Fig. S10c), water cluster distributions (Fig. S10c), sensitivities (Fig. S10d), and fragmentation

(Fig. 4) all suggest variable IMR conditions during this period of time. The IMR pressure regulator position, which was recorded for each mass spectra, was used as a higher temporal resolution proxy for the inlet flow rate, which was measured manually, as they share a linear relationship (Fig. S11). With a constant flow of 15 sccm of water vapor, the composition of gas in the IMR transitioned from 7% to 23% water vapor by volume. A non-exhaustive list of interdependent changes include: dilution, cluster ion distribution, and hydronium ion mobility.

- Dilution: the sample inlet flow was reduced by 75% relative to the initial flow and was diluted with water vapor by an additional 16% (absolute difference in dilution). The lower inlet flow rate resulted in enhanced dilution by water vapor and an increased signal of hydronium ions (Fig. S10c). The dilution of analytes and higher concentration of primary ions compete to decrease and increase sensitivities, respectively. Similar behavior may be expected as a function of altitude as the inlet flow rate depends on the pressure differential between ambient air and the IMR.

- Cluster ion distribution: the greater proportion of water vapor in the reactor also contributed to greater water cluster formation in the IMR (Fig. S10c). By lowering the average molecular weight of the IMR buffer gas, the additional water vapor reduced the effective temperature and collisional energy in the IMR, contributing to more water cluster formation. Proton affinities dictate whether a VOC will undergo proton transfer with hydronium-water clusters, partially explaining the difference in response between acetone and benzene (Fig. S10d). Additionally, the lower collisional energy reduced fragmentation (Fig. 4). Methyl ethyl ketone, acetonitrile, and acrylonitrile demonstrated declining quantitative ion fractions due to a greater abundance of analyte-water clusters. Benzene was the outlier since its quantitative ion fraction declined during this time while only the quantitative ion and $C_6H_7O^+$ had non-negligible contributions. The identity of this ion was unclear, but it was more abundant as the inlet flow rate decreased.

- Hydronium ion mobility: the hydronium ion mobility was reduced in the presence of the greater proportions of polar water molecules. The mixing ratio of water vapor in the IMR increased three-fold, increasing the average polarity of the buffer gas, and increasing the frequency of hydronium's ion-dipole interactions. Greater ion-molecule interactions and reaction cross sections reduced drift velocities and further reduced collision energy (Allers et al., 2020; de Gouw et al., 1997; Haber et al., 2004; Zhang et al., 2017). Calculated values of $k_{PTR}$ also vary with ion mobility such that nonpolar compounds, e.g., benzene, experience little to no change in $k_{PTR}$ while polar compounds, e.g., acetone, experience increasing $k_{PTR}$ with decreasing ion mobility (Chesnavich et al., 1980; Su, 1994; Su and Chesnavich, 1982).

The relative magnitude of these effects, likely among others, competed to influence the measured sensitivities. Deeper exploration into the effects of variable reactor conditions falls outside the scope of this study, but the consequences were apparent and considered when evaluating the PTR-DT in Section 4.1. Given the variable conditions, the evaluation is somewhat limited as some quantities, namely $k_{PTR}$, and $f$, are treated as constants in the toolkit, yet vary with the reactor conditions. This limitation is not unique to this study and should be considered. Regardless, the previous evaluation of the PTR-DT represents a practical and reasonable application as opposed to a best-case scenario, providing an assumed typical expected performance.

## 4.5 Instrument background measurements: limitations and comparisons

Each of the three clean air sources had limitations in this study. The HC trap was generally insufficient for scrubbing VOCs, likely due to extended use (~1 year) and reduced filtering capacity. In extreme cases, the measured HC trap measurements' signals exceeded ambient signals for some compounds such as methanol, acetaldehyde, acetonitrile, and monoterpenes. The

zero cylinder measurements also spanned the full measurement period, but lacked temporal resolution. A clear step change in signals was also observed for some ions when transferring from zero grade air to UHP nitrogen due to differences in contamination. The catalyst measurements were only available during the latter half of the measurements. Table S3 summarizes the average signals for each of these three clean air sources during the latter half of the measurement period for the standards presented in Tables S1 and S2.

### 4.5.1 Comparisons of clean air sources

To compare the three clean air sources, the average ratios of measured signals were compared for 815 high resolution ions (Fig. 6). Some ratios are not shown where the absolute difference between the average signals were less than 1 cps to avoid overinterpreting ratios of small numbers. Many high $m/Q$ ions were removed by this filter, indicating similar performance for the three clean air sources. Only the latter half of the measurements where the catalyst was available were considered. The zero cylinders, specifically the UHP nitrogen during this timeframe, and catalyst measurements were temporally adjacent, so they were compared one-to-one. To compare measurements made around the same time, the HC trap measurements were averaged into bins within ±4 h of the catalyst measurements. Ratios of unity indicate comparable performance while higher ratios indicate better performance by the catalyst.

Figure 6a shows that the catalyst outperformed the UHP nitrogen for the vast majority of ions. The UHP nitrogen performed better for 25 ions, most notably the ions typically associated with toluene ($C_7H_8H^+$), C8 aromatics ($C_8H_{10}H^+$), C9 aromatics ($C_9H_{12}H^+$), and naphthalene ($C_{10}H_8H^+$). Investigations focused on low concentrations of these aromatics may benefit from using UHP nitrogen as opposed to a catalyst if a difference of 6–44 cps (several pptv in this study) is significant. These ratios are likely to vary across cylinders while a catalyst is likely to be more consistent provided it is not overloaded nor degraded.

The catalyst outperformed the HC trap for all but 18 ions, although the differences were negligible for these 18 ions (Fig. 6b). The top 7 ratios were observed for halogenated fragmentation products, possibly indicating reducing filtering capacity of long-lived halocarbons. This comparison was not wholly fair as the HC trap had been in use for a year. Further analysis with a newer trap is necessary to assess its best-case performance as well as determine its lifetime before filtering capacity is diminished for key species.

From these comparisons, the catalyst performs the best for the widest range of compounds. The catalyst did not produce significant amounts of oxygenated VOCs relative to the other clean air sources. The HC trap, despite a year of constant use, performs second best. To reiterate, a newer HC trap is expected to have better performance. However, the instrument background signals for common laboratory VOCs, e.g., solvents and monoterpenes, should be closely monitored for saturation of the trap. While the UHP nitrogen performed the worst for most ions, contamination is likely to vary significantly from one cylinder to another. Also, a reduced abundance of charge transfer products was not observed in the absence of oxygen when using the UHP nitrogen. Notably, the $O_2^+$ and $NO^+$ signals were not different between the HC trap, catalyst, UHP nitrogen, and ambient measurements. This study does not seem to indicate any strong trends regarding functional groups and the catalyst

does not seem to produce oxygenated species. Individual use cases may benefit from different methods or combinations of methods in sequence. Periodic measurements of different clean air sources may serve to validate the frequent measurements, the HC trap measurements here, or serve as a reference for corrections.

### 4.5.2 Limits of detection

Average limits of detection (LODs; 5 s) were determined using the catalyst instrument background measurements and the measured or estimated sensitivities. Tables S1 and S2 list the standards' average LODs and Fig. 7 shows the LODs for the 615 quantified species including the standards. At higher masses with lower instrument background signals, the LODs converged to 0.1±0.3 pptv per an exponential fit. However, many of these species were only *semi*-quantified due to unconstrained transmission and detection efficiency in the high *m/Q* regime (>300 Th) and undetermined fragmentation, leading to

underestimated LODs. D3, D4, and D5 siloxanes ($C_6H_{18}O_3Si_3$, $C_8H_{24}O_4Si_4$, and $C_{10}H_{30}O_5Si_5$, respectively) were directly calibrated and suggest the LOD limit for the instrument was in the single pptv range for 5 s averaging. These siloxanes can be relatively abundant in the atmosphere due to volatile chemical products (Coggon et al., 2018) and tended to persist inside the instrument following calibrations, causing higher instrument background signals and LODs. Such persistent effects from calibrations should be considered when targeting species at the single pptv level in the ambient atmosphere. The highest LODs

were observed for common laboratory solvents, e.g., methanol and ethanol, and other abundant trace gasses, e.g., acetaldehyde and acetic acid, which may not be fully removed by the catalyst, have sources within the instrument, and/or are produced by the ion source.

### 4.6 Applications to low-signal measurements

### 4.6.1 Cooking emissions

Cooking emissions include VOCs that contribute to secondary organic aerosol (Takhar et al., 2021) and ozone formation potential (Cheng et al., 2016). Aldehydes are commonly emitted while cooking using different methods, oils, and foods (Atamaleki et al., 2022; Liang et al., 2022; Schauer et al., 1999). Here, furfural is used as a tracer for gas-phase cooking emissions and is formed from sugar-degradation reactions (Kroh, 1994). Furfural is also commonly associated with biomass burning (Gilman et al., 2015), but acetonitrile, which is another common tracer for biomass burning (Coggon et al., 2016), did

not exhibit similar enhancements. The sampling location in this study was in close proximity to the main university food court and oxygenate-rich plumes were observed around noon during the university lunch rush (Fig. 8a–b).

These oxygenated VOCs are attributed primarily to aldehydes during these cooking episodes. The $C_8H_{16}OH^+$ and $C_9H_{18}OH^+$ chromatograms during these plumes were each dominated by a single peak and demonstrated significant fragmentation to the dehydration product, which is common of aldehydes in PTR-MS (Buhr et al., 2002; Pagonis et al., 2019).

For the purposes of analyzing these cooking emissions, the real-time signals of $C_8H_{16}OH^+$ and $C_9H_{18}OH^+$ were calibrated as the isomers of octanal and nonanal, respectively, using the PTR-DT. The PTR rate constant was derived using estimated

polarizability and permanent dipole moment (Sekimoto et al., 2017) where all aldehydes and ketones of the same molecular formula have the same values. Fragmentation was derived empirically as described in Section 3.1.1.

Lower limits of detection allowed for the identification and quantification of larger molecular weight species and their contributions to cooking emissions. For example, the signal of $C_{20}H_{40}OH^+$ was above the limit of detection during these cooking plumes (Fig. 8b). Although this signal was not resolved on the GC, $C_{20}H_{40}O$ is attributed to the isomers of icosanal due to its association with these cooking plumes. Fragmentation could not be determined, but that of nonanal was assumed. Icosanal's fragmentation was likely greater, so the reported concentrations are semi-quantitative and underestimated. The signal for $C_{15}H_{30}OH^+$, attributed to isomers of pentadecanal, was similarly quantified.

Emission ratios of these aldehydes relative to furfural were derived for four cooking plumes (April 1–4; 11:00–13:00; Fig. 8c–d) using 1 min averaged data. The measurements suggest average emission ratios of ~210, ~180, ~12, and ~1.8 pptv ppbv$^{-1}$ furfural for octanal, nonanal, pentadecanal, and icosanal, respectively. There was large day-to-day variability (e.g., octanal ranged from 170–290 pptv ppbv$^{-1}$) as different cooking methods, temperatures, oils, and foods yield different emission rates (Klein et al., 2016; Peng et al., 2017; Song et al., 2022). Schauer et al. (1999) report meat charbroiling emission ratios 700 for octanal, nonanal, and pentadecanal as ~1340, ~1170, and ~70 pptv ppbv$^{-1}$ furfural, respectively, or ~6 times the values in this study, although with a similar distribution. They sampled directly downstream of a charbroiler exhaust which included a filter and grease extractor with an expected particle mass removal efficiency of 60% (Schauer et al., 1999). The measurements in this study were made downwind of the exhaust of many diverse sources and after additional losses to surfaces in the remaining ventilation system, cooling and subsequent partitioning into the aerosol phase, and losses to surfaces in the sample 705 line and Vocus inlet. Due to the assumed close proximity to the emission source, photochemical losses are assumed negligible. Icosanal is expected to have a saturation vapor concentration (C*) of ~1×10$^{-1}$ µg m$^{-3}$ as estimated using SIMPOL.1 (Pankow and Asher, 2008) and partition >99% to the aerosol phase assuming at least 10 µg m$^{-3}$ organic aerosol (Donahue et al., 2006; Pankow, 1994). This distribution speaks to the significant total emissions of icosanal given its detection in the gas phase.

Additional plumes were noted in the evening ~18:00–21:00 local time (Fig. 8a–b) and are attributed to the dinnertime 710 rush. Furfural, octanal, and nonanal demonstrated enhancements during these times, but pentadecanal and icosanal did not. During lunch rushes, furfural demonstrated significant ambient variability which indicated relatively little mixing and a nearby emission source. The evening furfural plumes did not show such variability which suggests more time to mix, cool, and partition to aerosols. As such, the concentrations of the lower volatility aldehydes fell below the limit of detection. Notably, the main university food court is less busy in the evening as several kitchens stop serving ~17:30.

**4.6.2 Cyclic volatile methyl siloxanes**

Volatile chemical products (VCPs) have been growing in importance as petrochemical air pollutants and contributors to secondary pollutants in urban environments (Coggon et al., 2021; Gkatzelis et al., 2021b; McDonald et al., 2018). Gkatzelis et al. (2021a) identified two cyclic volatile methyl siloxanes as tracers for VCPs: D4 and D5 siloxane. PTR-MS signals for the D3–D5 siloxanes' elemental compositions are typically assumed to be solely from these isomers. Chromatograms in this study,

aided by high PTR sensitivity, support this assumption as no additional isomers were detected (Fig. 9a). These siloxanes are expected to react slowly with hydroxyl radicals with lifetimes of days (Alton and Browne, 2020) and the resulting first generation oxidation products are expected to exist primarily in the gas phase (Alton and Browne, 2022), although they were not detected in this study. The time series for these siloxanes show broad, regional enhancements in the morning, coinciding with the morning commute, and other brief, mid-day enhancements from more local VCP sources, typically on the order of a few pptv for D3 and D4 siloxanes (Fig. 9b).

### 4.6.3 Organosulfur compounds

Some organosulfur compounds were detected during these measurements: dimethyl sulfide (DMS), dimethyl disulfide (DMDS), and methanethiol. These compounds have various emission sources such as the degradation of dimethylsulphoniopropionate released by some phytoplankton (Carpenter et al., 2012), biomass burning (Koss et al., 2018; Meinardi et al., 2003), wastewater (Glindemann et al., 2006), and industry (Texier et al., 2004; Toda et al., 2010). Organosulfur compounds are often added to natural gas as odorants to help detect leaks. These compounds represent unique elemental compositions and may serve as useful tracers for various emission sources.

Chromatograms for $C_2H_6S_2$ yielded a single peak. This formula was identified as DMDS based on its fragmentation pattern with only the loss of a methyl group, as observed elsewhere (Perraud et al., 2016). DMDS's polarizability and permanent dipole moment (Cappellin et al., 2010) were used to estimate its $k_{PTR}$. Methanethiol was quantified similarly while DMS was a standard. DMS was also associated with a single chromatographic peak. Methanethiol was not resolved on the GC, so its fragmentation could not be determined, and the reported mixing ratios are lower bounds.

Figure 10 summarizes the observations of organosulfur compounds including their temporal relation to methane. Methane and sulfur compounds share some common sources in agriculture and consumer gas leaks, and these scatter plots may give some insight into the use of the sulfur compounds as tracers for these sources. Methanethiol tended to correlate better with DMS than methane. Moreover, methanethiol strongly correlated with DMS for nearly the entire measurement period ($R^2$ = 0.79), indicating common sources. Some of the variability in their correlation may be attributed to chemistry as methanethiol reacts ~8 times faster than DMS with hydroxyl radicals during the day (Wine et al., 1981).

Sub-pptv enhancements in DMDS coincided with enhancements in methane and DMS during separate plumes. Specifically, DMDS seems to have at least two distinct sources, one of which is separate from these other organosulfur compounds. DMDS may not serve well as a tracer over long transport times due its fast reaction rate with hydroxyl radicals, $k_{OH}$, of $2\times10^{-10}$ cm$^3$ molec$^{-1}$ s$^{-1}$ at 298 K (Tyndall and Ravishankara, 1991; Wine et al., 1981). This quick removal is apparent as the presence of methane tended to extend beyond that of DMDS in the observed plumes, indicating local emissions. Regardless, PTR-MS observations of ambient DMDS are sparse due to previous instrumental limitations.

# 5 Conclusions

VOCs were measured via a Vocus PTR-TOF-MS in Boulder, Colorado in spring 2021. During these measurements, three different clean air sources were used to determine instrument background signals. Fast calibrations were done on a 2 h timescale in addition to multipoint calibrations done daily. A toolkit was developed to capitalize on these frequent calibrations to estimate sensitivities of VOCs which were not directly calibrated on the same 2 h timescale and to assess the instrument's performance. Finally, a few applications of quantified low-signal species were demonstrated including cooking emissions, cyclic volatile methyl siloxanes, and organosulfur compounds.

The PTR-DT served as a tool to monitor instrument performance and to rapidly calibrate species which do not have standards using simple reaction kinetics on the same timescales of routine calibrations during the Boulder measurements. This tool is based on general PTR-MS operational principles and is applicable to any such measurements, but the code is available to be adapted to different situations as necessary. Similar reactor conditions and ion chemistries allowed for the best intercomparisons between different times and the best performance by the toolkit. Monitoring the output fitting parameters allowed for identification of varying instrument behavior, e.g., varying reactor conditions, and provided insight into the effects on sensitivities. Standards' sensitivities estimated using the PTR-DT tended to skew toward positive residuals when compared to the corresponding measured sensitivities due to factors such as unconstrained fragmentation, unconstrained transmission, and non-constant ion chemistry. Although there are limitations in some of the underlying assumptions, the PTR-DT recreated the standards' sensitivities within 1±8% on average.

Fast, frequent calibrations were used to capture short-term variability in sensitivity as opposed to applying a normalization correction. Multipoint calibrations were used to assess accuracy and the fast calibrations were 5±6% lower than the multipoint calibrations. This discrepancy is attributed to an offset between the instrument background and calibration measurements regardless of clean air source. This offset was caused, in part, by incomplete mixing of the standard with diluent.

Over the course of the measurements, the sample inlet was obstructed. The resulting reduction in sample flow prompted increased analyte dilution, greater water cluster formation, and reduced ion mobility in the IMR. These effects contributed to variable sensitivities with different responses for different analytes. Similarly, fragmentation rates varied with the greatest variability attributed to isoprene (6% relative standard deviation). Due to these changing IMR conditions, traditional normalization against the primary ion signal failed to account for the variability in instrument response, hence the use of fast, frequent calibrations. The sample inlet obstruction was not unique to this study or instrument and IMR conditions should be closely monitored to maintain consistent ion chemistry.

Comparisons of the three clean air sources (HC trap, UHP nitrogen, and catalyst) found that the catalyst yielded the lowest instrument background signals for the vast majority of ions. The UHP nitrogen outperformed the catalyst for a select few, yet commonly quantified, species (toluene, C8 aromatics, C9 aromatics, and naphthalene) by a narrow margin of several pptv. The catalyst performed better than the HC trap for all but a few ions, but their performances were comparable for these exceptions. Notably, no trends in elemental composition, specifically oxygenated species, were observed in the catalyst

measurements. With the catalyst instrument background measurements, LODs were in the range of a few pptv at a 5 s measurement interval for the majority of measured standards.


*Code availability.* The code for the PTR-DT is available on the de Gouw group webpage: https://sites.google.com/view/de-gouw-lab/instruments/ptr-data-toolkit?authuser=0&pli=1.

*Data availability.* The data used in this study have been made available in the Open Science Framework:
https://doi.org/10.17605/OSF.IO/KZPEV.

*Author contributions.* ARJ performed the VOC measurements and subsequent analysis as well as developed the PTR-DT. RBH performed the analysis of cooking emissions. ARK provided insights into the instrumentation. JdG provided guidance throughout the study.


*Competing interests.* ARK is a scientist employed by Tofwerk AG, which commercially produces the mass spectrometer used in this study.

*Acknowledgements.* We would like to thank Jose L. Jimenez and Anne Handschy for providing the additional trace gas
measurements. We would like to thank Brian Lerner and Megan Claflin for their help in preparing for and interpreting the GC measurements.

*Financial support.* This study was supported by the Cooperative Institute for Research in the Environmental Sciences (CIRES) Graduate Research Award.

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

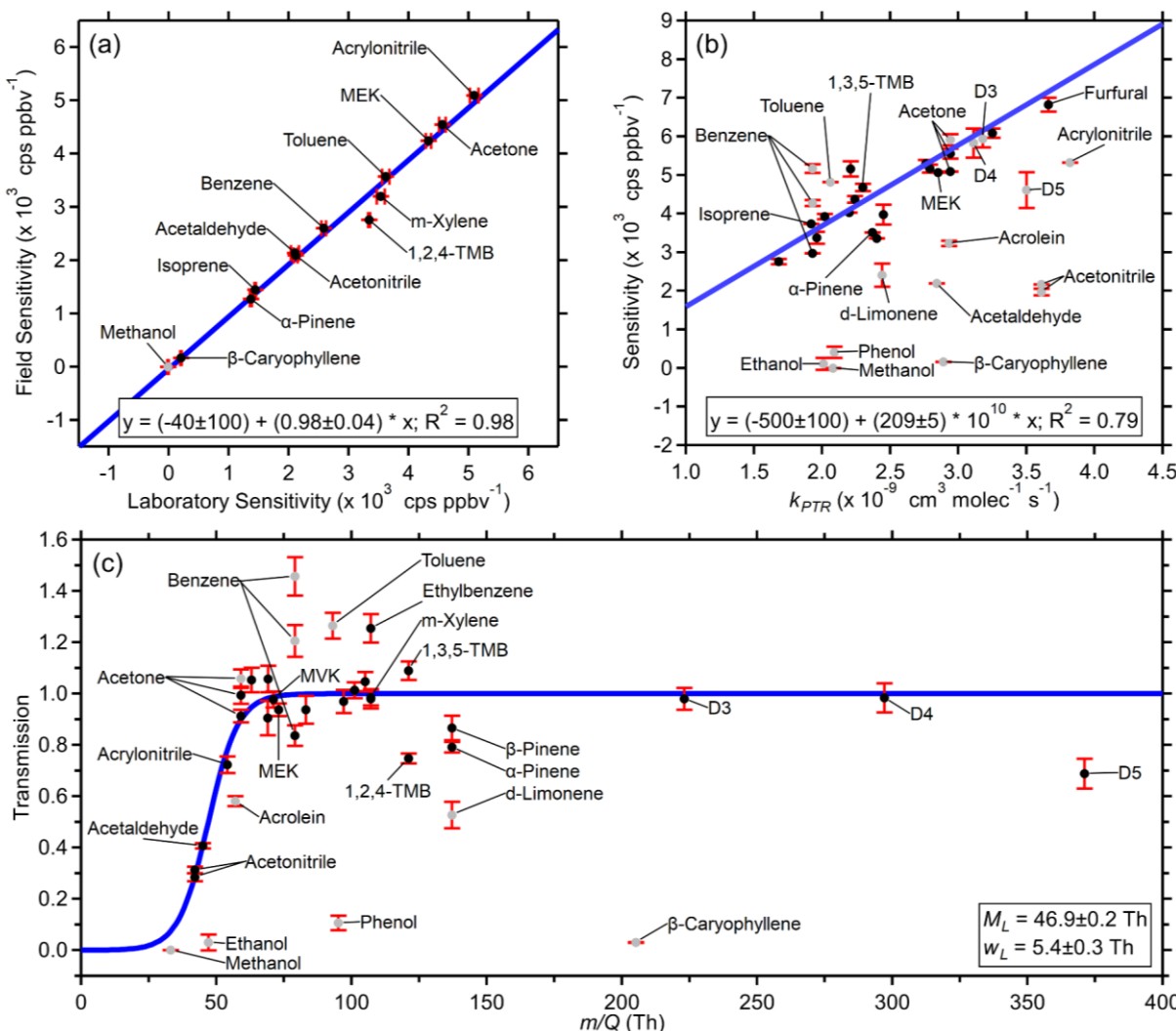

**Figure 1: Example fits from the PTR-DT including (a) an orthogonal distance regression of the first field calibration sensitivities and the pre-field laboratory calibration sensitivities, (b) an orthogonal distance regression of the first field calibration sensitivities and the respective PTR rate constants, and (c) an *m/Q*-dependent transmission curve for the first field calibration. Uncertainties in laboratory sensitivities are the standard deviation of replicate measurements. Field-estimated sensitivity uncertainties of laboratory standards were propagated with the uncertainties of the regressions in step B. Transmission uncertainties were propagated from previous steps. Grayed-out standards were excluded from the respective fits as described in Section 3. Figure S7 shows similar fits using the post-field laboratory calibrations.**

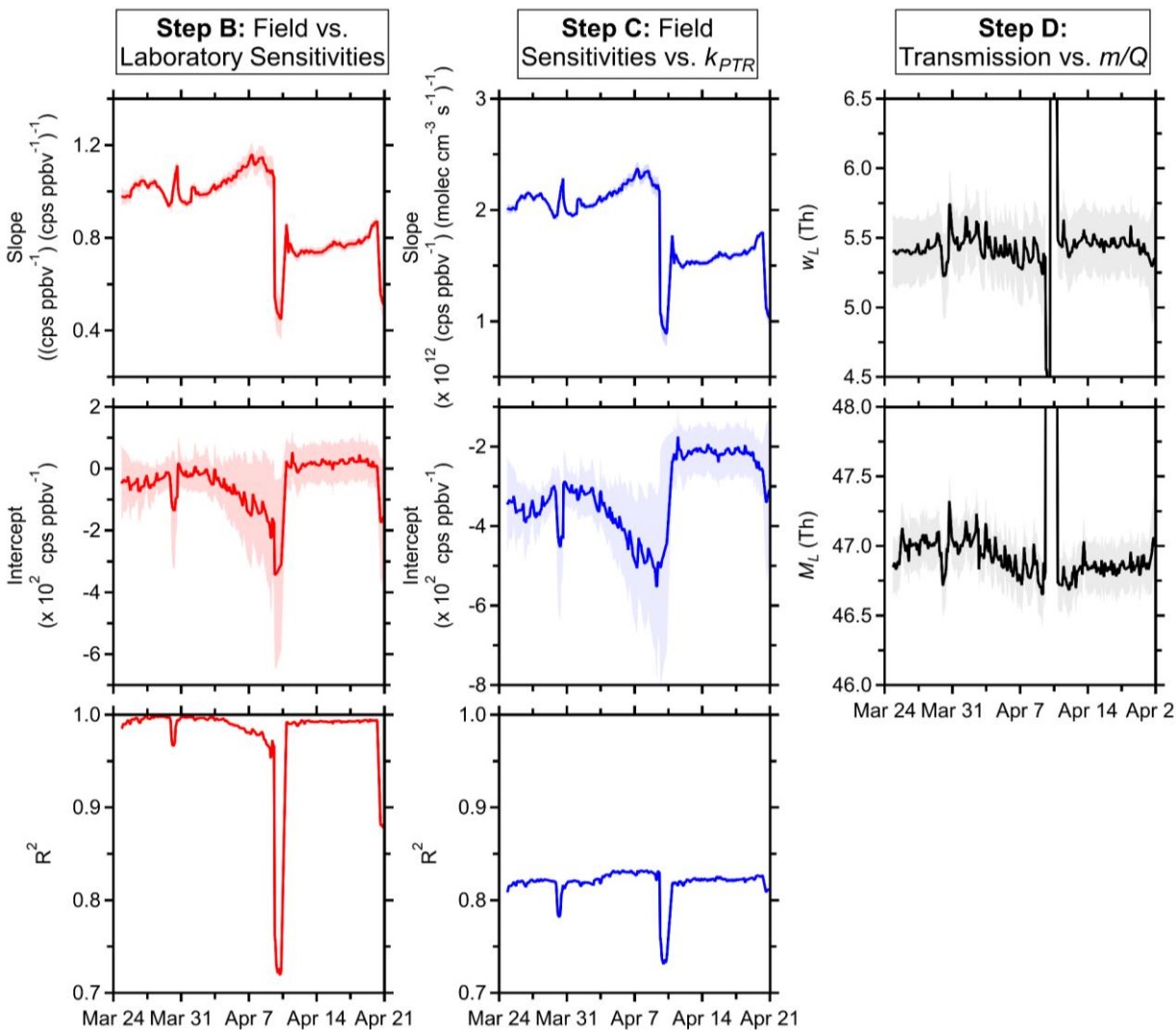

**Figure 2: Time series of PTR-DT fitting parameters derived using the standards in Tables S1 and S2 with the exceptions outlined in Section 3. Shaded regions represent uncertainties in the fitting parameters. Scales for the step D plots are truncated to show detail, but only periods where the ion source malfunctioned are not shown.**

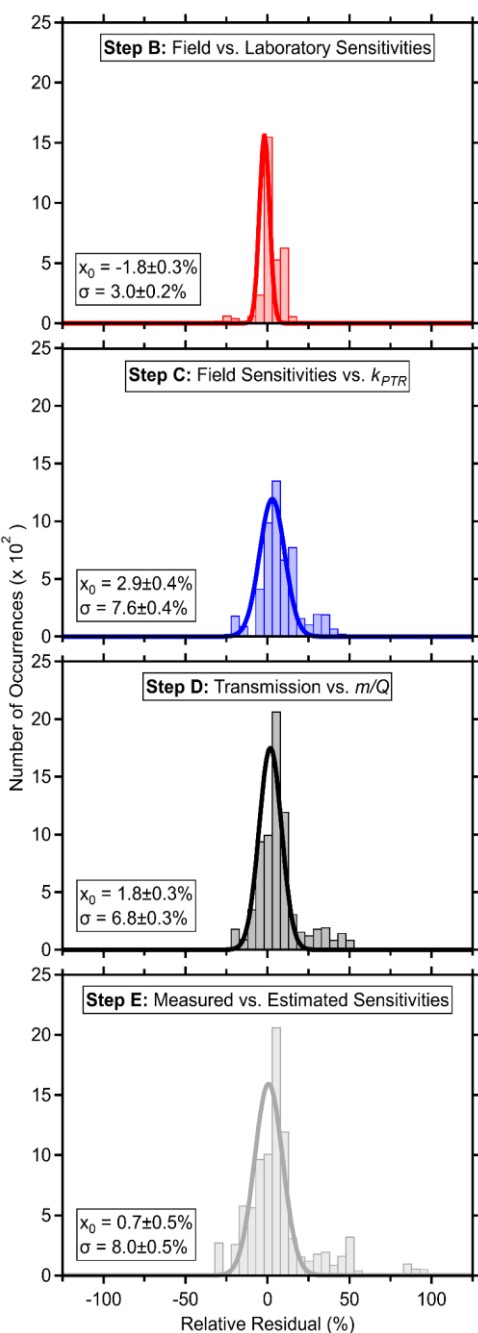

**Figure 3: Relative residual histograms (5% bins) with fits to a normal distribution (average, $x_0$ and standard deviation, $\sigma$, provided with fitting uncertainties) comparing standards' measured field and laboratory sensitivities (B), measured (or field-estimated in the case of laboratory standards) and fit sensitivities (C), measured and fit transmission (D), as well as measured sensitivities and those calculated using the parameters and regressions from the PTR-DT (E) for each fast calibration. Field-estimated sensitivities of laboratory-only standards were derived from the pre-field laboratory sensitivities. Residuals were defined as the difference between the fit and measured or derived value over the measured or derived value. Standards were excluded if fragmentation rates were not determined. Periods where the ion source malfunctioned were also excluded. Fits assume a baseline offset of 0.**


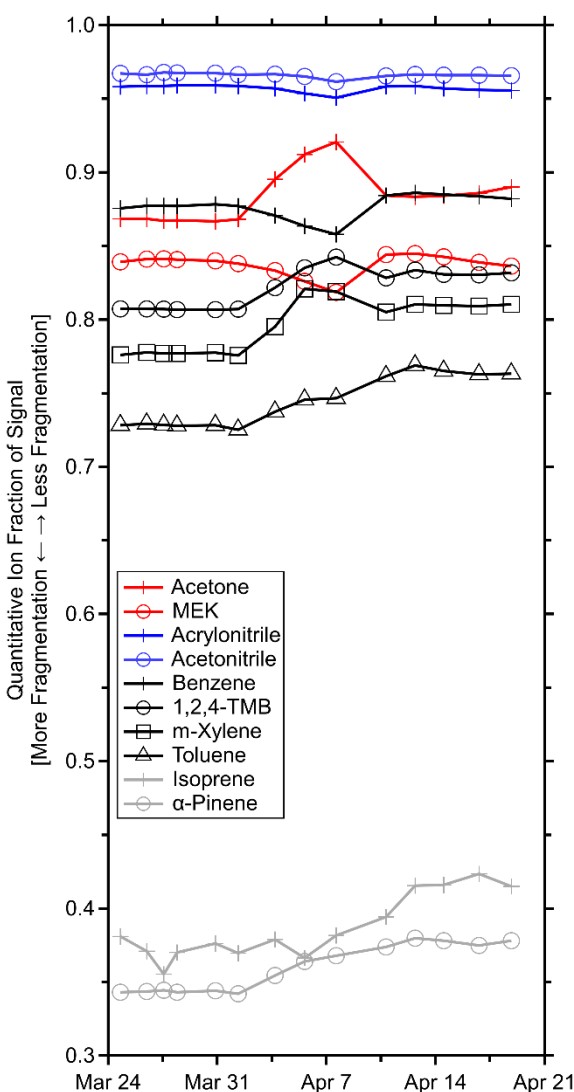


**Figure 4: Time series of quantitative ion fractions measured during the GC field calibrations for the standards in Table S1. Methanol, acetaldehyde, and β-caryophyllene are omitted as they were not resolved on the GC column and fragmentation could not be determined. Isoprene was corrected for transmission.**

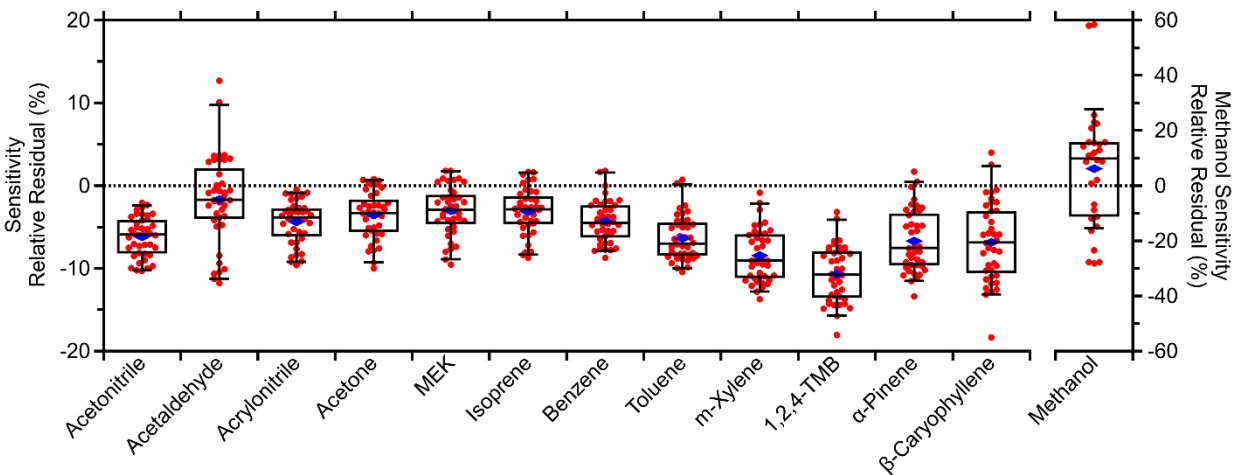

**Figure 5: Relative residual box plots comparing field standards' (Table S1) sensitivities from each multipoint calibration against adjacent (±4 h) fast calibrations. Relative residuals were calculated as the ratio of their difference to the multipoint sensitivity. That is, a relative residual of -10% means the fast calibration was 10% lower than the corresponding multipoint calibration. Four methanol multipoint calibrations at the beginning of the field measurements were excluded due to erratic behavior (e.g., negative sensitivities). Boxes and whiskers represent the 5th, 25th, 50th, 75th, and 95th percentiles. Blue diamonds represent the mean relative**
**residual for each standard.**

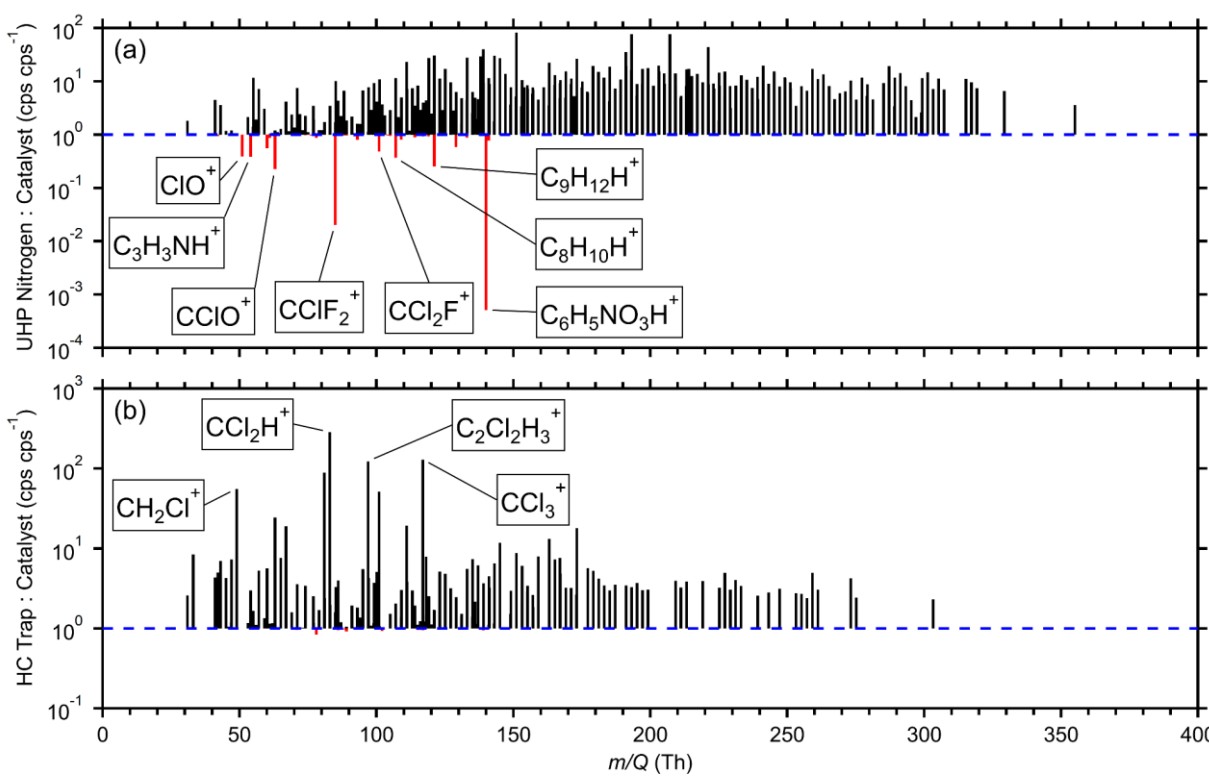

**Figure 6: Average signal ratios for 815 high resolution ions comparing (a) the UHP nitrogen against the catalyst during the latter half of the field measurements, and (b) the HC trap against the catalyst during the latter half of the field measurements. The UHP nitrogen and catalyst measurements were temporally compared one-to-one. HC trap measurements were averaged into bins within ±4 h of each catalyst measurement. Ratios were excluded where the absolute difference between the average signals was less than 1 cps. For ratios where the catalyst performed better, the trace is black. For ratios where the UHP nitrogen (a) or HC trap (b) performed better, the trace is red.**


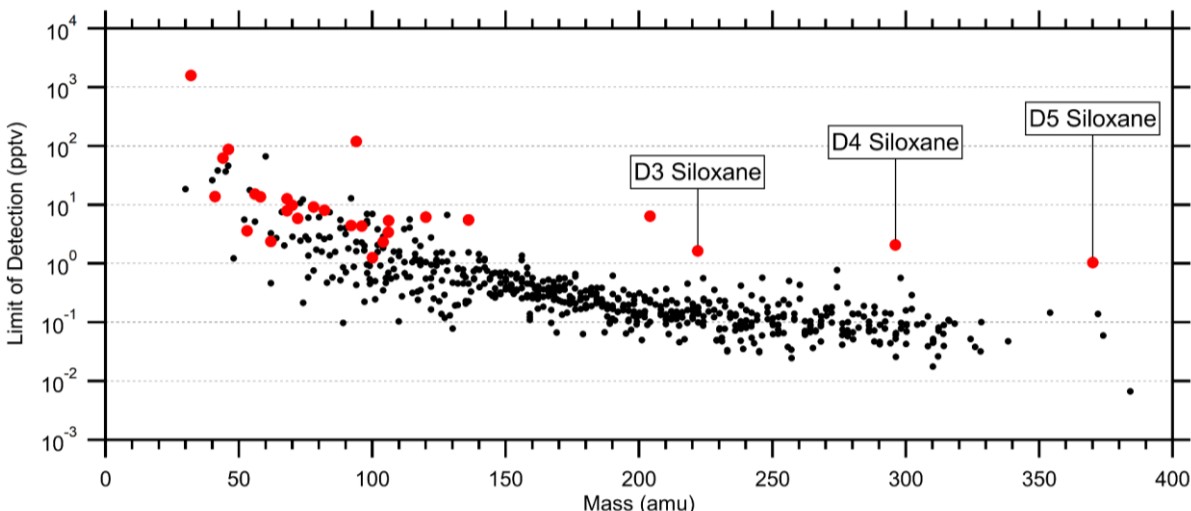

**Figure 7: Average limits of detection (LODs; 5 s) for 616 quantified species using catalyst instrument background measurements. LODs were calculated as three times the standard deviation of the instrument background divided by the sensitivity. Red markers indicate standards (Tables S1 and S2). LODs at high mass are biased low due to unconstrained fragmentation while the siloxane standards provide a more realistic LOD limit.**

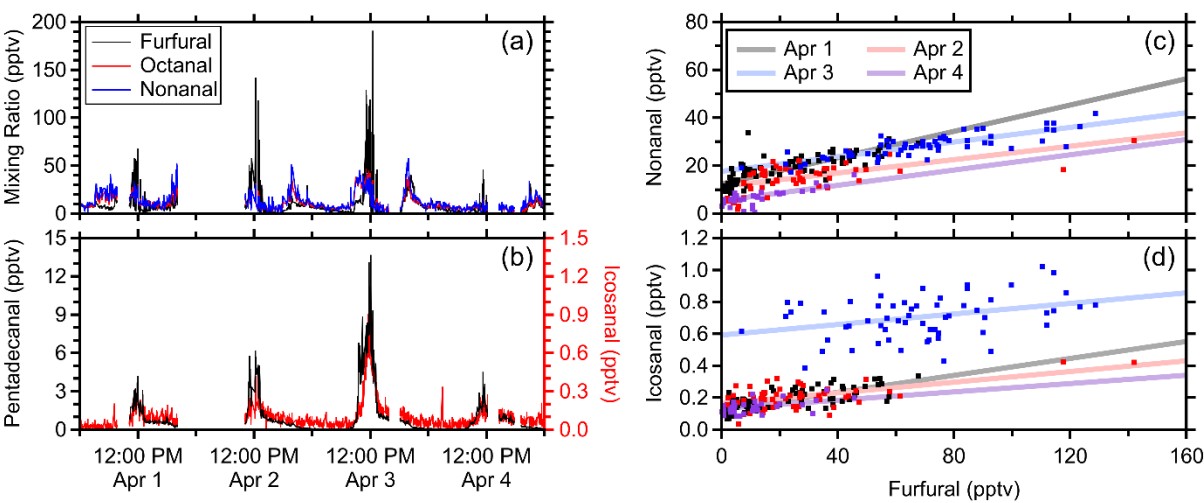


**Figure 8: Aldehyde plumes during four lunchtime cooking episodes (a–b) and orthogonal distance regressions of nonanal (c) and icosanal (d) against furfural during these episodes from 11:00–13:00. All data are averaged to 1 min timescales.**

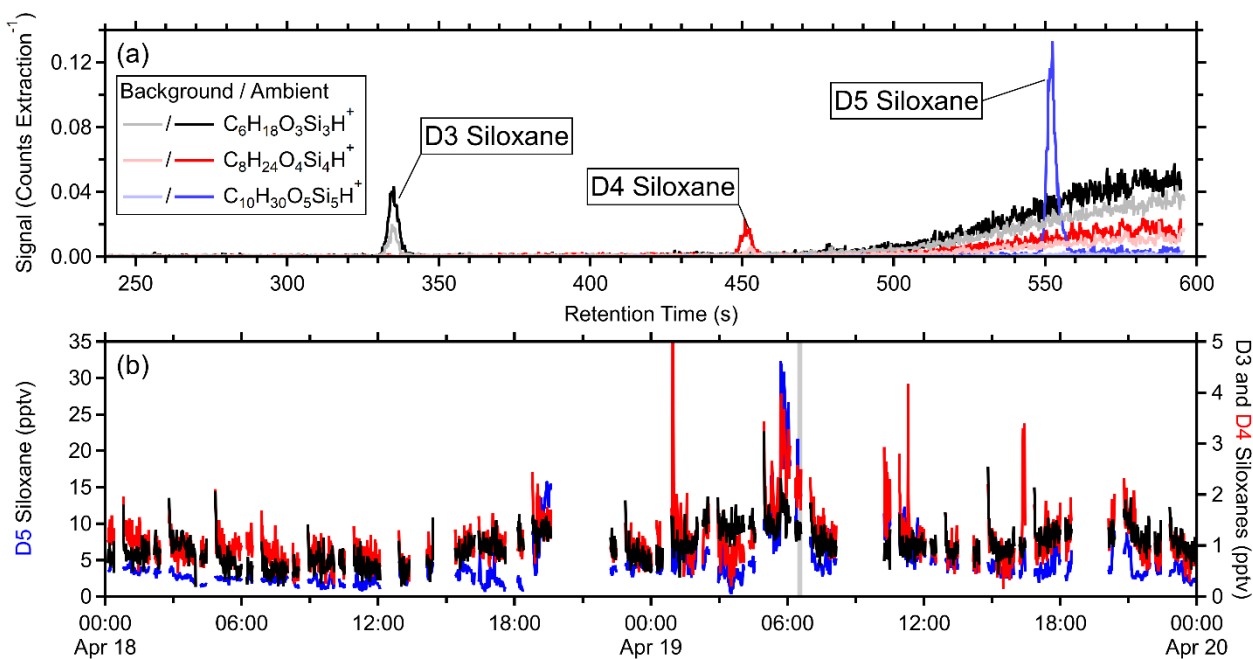

**Figure 9: (a) Ambient (April 19, 06:25–06:35) and instrument background (April 18, 21:47–21:57; lighter traces) chromatograms of the parent ions typically associated with D3–D5 siloxanes and (b) time series of these siloxanes (averaged to 1 min). The gray, shaded region in (b) represents the GC sample collection time.**

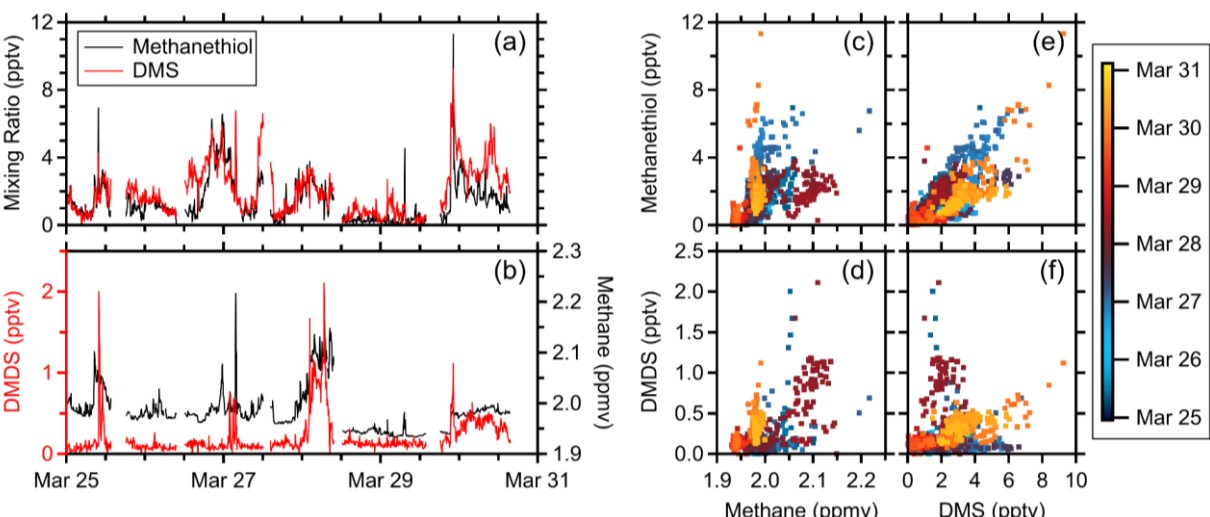

**Figure 10: Time series (a–b) and scatter plots (c–f) of observed organosulfur species and methane. Scatterplots include the same range of data as the time series. All data are averaged over a 5 min timescale.**
