# Peer review of "Measurements of VOCs in ambient air by GC- and Real-Time-Vocus PTR-TOF-MS: calibrations, instrument background corrections, and introducing a PTR Data Toolkit"

_EGUsphere, 2023_

## Author Comment (AC1)

**Response to Referee #1's Comments for: "Measurements of VOCs in ambient air by Vocus PTR-TOF-MS: calibrations, instrument background corrections, and introducing a PTR Data Toolkit"**

We thank the Editor and Referees for their time and constructive feedback on the manuscript. We have addressed all referee comments and updated our manuscript accordingly. Please find our summarized responses below. Referee comments are printed in **bold black**, our specific replies are in this blue color, quotes from the submitted manuscript are *in this blue color and italics* and quotes from the revised manuscript are provided in this *green color and italics*.

In addition to corrections made in response to Referee comments, we have made additional corrections for clarity and grammar.

We have corrected methanol's "Quantitative Ion" entry in Table S1 to reflect that the sensitivities and LODs correspond to protonated methanol summed with its water cluster (as already stated in the text, line 167).

Following discussions and troubleshooting with members of the community, we have fixed a bug in the PTR-DT which prevented it from compiling in Igor Pro 8. We have clarified which versions of Igor Pro are known to work with the PTR-DT.

- (line 200) *"Currently, version 1.1 is known to function in Igor Pro 8 and 9."*

We identified minor mistakes in Figure 6 where traces were misnamed. The scientific conclusions have not changed aside from the minor observation that *acetonitrile* experienced reduced fragmentation due to increased water clustering as opposed to *acetone* (line 600).

**Anonymous Referee #1 Responses:**

**Jensen et al provide a comprehensive and useful analysis of best practices for interpretation of high-resolution PTR data. The manuscript provides a detailed discussion of factors contributing to PTR sensitivity and its variability in the field. The manuscript will be a helpful asset to the community and should be published following the authors attention to the following comments:**

We thank Referee #1 for the time to evaluate this work and for their detailed feedback to help improve this manuscript.

**General Comment:**

**The authors do an excellent job discussing how fragmentation of a parent ion impacts its sensitivity. That is, fragmentation reduces the expected signal at MH+ as some fraction of the molecules fragment to smaller ions. This impacts the retrieved sensitivity and the comparison of the measured and expected sensitivity. The value of f for a molecule can be determined from the GC. There is less discussion about the positive bias that fragmentation can invoke. For example, at 69 m/Q (or the exact mass of C5H9+), some fraction of the ions detected here are protonated isoprene (you know this fraction from your C5H9+ chromatogram) and some fraction is fragmented larger molecules. This can be significant. From what I can tell the toolbox here does not address this issue of fragmentation. I appreciate that this is tricky. If the authors do not want to tackle this, I think that is fine, but it would be helpful to provide a short statement about how this could impact this analysis.**

We agree that contributions to target ion signals from the fragmentation of larger ions are a significant issue worthy of discussion. Referee #1 is correct that the toolkit does not address this issue; that is, the sensitivities calculated by the toolkit apply to the signals of the target analyte alone. Corrections for such biases have been used in literature, e.g., Vermeuel et al. (2023) corrected for aldehyde fragmentation contributions to isoprene. However, in our own experience, we have found that methods to apply these corrections will vary significantly between analytes. Regardless, we have added a brief discussion to Section 4.1.6:

- (line 471) *"Additionally, the PTR-DT does not account for spectral interference. That is, the fragmentation or adduct formation of other species increase the measured signals of a target analyte. Sensitivities from the PTR-DT, which correspond to the target analyte alone, will yield overestimated concentrations. Values of $k_{PTR}$ used in the PTR-DT will also only correspond to the target analyte and have no relation to interfering fragment ions. However, these limitations are not unique to the PTR-DT and also apply to the use of standards to measure sensitivities. To account for these interferences, analyte- and interference-specific corrections could possibly be applied to the estimated sensitivities, but these interferences may be on shorter timescales than routine calibrations. Instead, corrections informed by GC may be applied to the real-time measurements as demonstrated by Vermeuel et al. (2023) for aldehyde fragmentation contributions to isoprene's quantitative ion. Briefly, they used GC to characterize the relative abundance of $C_5H_9^+$ (the quantitative ion used for isoprene) compared to the parent ions for n-aldehydes. Then, they scaled the real-time signal for those aldehydes by that relative abundance*

*and subtracted those contributions from the real-time signal for $C_5H_9^+$. The remaining signal uniquely corresponded to isoprene and was calibrated using the isoprene sensitivity."*

The above edit to the manuscript is also meant to address some specific comments related to this general comment.

**Specific Comments:**

**Line 115: Please confirm whether the entire inlet or just the Vocus subsampling inlet was overflowed for calibration and zero.**

We clarified the sample flow path to help address this point and also modified our inlet schematic (now Fig. S1) to include the flow path used in this study:

- (line 94) *"Air was drawn via an external pump connected to the Vocus inlet such that the sample line led directly into the Vocus inlet for subsampling and the excess flow was removed from a perpendicular line."* Has been updated to:
- (line 110) *"Air was drawn via an external pump connected to the Vocus inlet such that the sample line led directly into the Vocus inlet (sample flow directed toward the IMR) for subsampling and the excess flow was removed toward the external pump via a perpendicular line also attached to the Vocus inlet (Fig. S1a)."*

Only the subsampling line was overflowed:

- (line 138) *"Excess flow was drawn downstream to the external pump (Fig. S1a) and the main sample line upstream of the Vocus inlet was unaffected aside from reduced flow rates of ambient air (at most, a reduction of ~0.3 L min$^{-1}$ at STP)."*
- (line 156) *"Fast calibrations were performed every 2 h by overflowing the Vocus inlet (as described for instrument background measurements)"*

**Section 2.3: Please confirm if the inlet for the HC trap and the catalytic zero source were drawn from room air or from ambient air.**

We have included this information:

- (line 145) *"The inlets for both the HC trap and catalyst drew from room air."*

**Line 175: This equation (E3) holds, so long as another (larger) molecule does not fragment into the detected ion [RH+]. I agree that E3 is correct in isolation, but in the atmosphere if a large fraction of the measured signal at RH+ is not from R but from a larger molecule that fragments, the sensitivity could not be applied to [RH+] to deduce [R] without knowing the fraction of the signal at [RH+] that is from R. Take for example isoprene, only 40% of isoprene is retrieved at RH+ (per your table S1), but the signal at RH+ is comprised on many other molecules beyond isoprene. This could be extracted from the chromatogram as well for the ambient data. Perhaps I missed it, but how is this side of fragmentation being accounted for?**

We agree that Eqn. 3 is a simplification which does not necessarily hold when measuring the complex atmosphere. Our responses to Referee #1's general comment and their later comment on "spectral interference" apply here. We do not account for such contributions from fragmentation. The sensitivity estimates could possibly be modified to account for these issues, but the dependence would likely be on shorter timescale than calibrations (temporary plumes). Instead, we recommend correcting the measured signal to approximate the isolated analyte, then apply the estimated sensitivity.

We attempted to explain that Eqn. (3) makes assumptions by:

- (line 177) "*Assuming no additional, outside factors, e.g., passivation effects and spectral interference, then $S_{inst}$ is expected to equal the measured sensitivity, $S_{meas}$.*"

However, we believe the clarification of "spectral interference" as suggested by Referee #1 improved this point. Additionally, we have included:

- (line 221) "*Equation (3) is a simplification since atmospheric measurements are complex and interferences are common. The PTR-DT does not account for spectral interference as discussed in Section 4.1.6.*"

**Line 200: These are Tables S1 and S2, not Tables 1 and 2 (took me a while to find them).**

Thank you for identifying this mistake (and apologies for the confusion). We have updated them to Tables S1 and S2 (same for Fig. 7's caption).

**Line 205: I'm a bit confused by this sentence. Why does it matter if the transmission function is different for the fragments. Is this because you need to know T(m/Q) to accurately determine f (i.e. if the transmission of the fragment is not accounted for and it is smaller, you would overpredict the actual value of f?) Otherwise, isn't the value of T(m/Q) in equation 3 specific to RH+? Sorry, if I'm turned around on this a reader may be as well, so it wouldn't hurt adding a sentence or two here to more fully describe this.**

In calculating $f$, we hope to account for all ions produced in the IMR (in the absence of T($m/Q$)) from some parent ion, RH+. The fragments once belonged to RH+ and represent some of the analyte that has undergone ionization, but the BSQ influences the measurement after fragmentation. A lower transmission efficiency causes undercounting of those fragments and overestimates the relative abundance of the quantitative ion. We have attempted to add clarification:

- (line 255) *"Values of f should reflect the product ion distribution in the IMR rather than the measured distribution. Without accounting for transmission efficiency for these fragments, the sum of all ions produced by a standard's ionization would be underestimated and f as well as the calculated sensitivity would be overestimated."*

**Line 215: It would be interesting to add how many k(PTR) values are known, calculated, vs estimated based on parameterization.**

In this analysis, all values of $k_{PTR}$ were estimated based on parameterization:

- (line 226) *"In this study, all values of $k_{PTR}$ were calculated based on the reactor conditions as well as molecular polarizability and permanent dipole moments from the literature, if available, or otherwise estimated based on Sekimoto et al. 's (2017) parameterizations."*

We relied on this parameterization due to a lack of experimental and calculated values $k_{PTR}$, particularly for our operating conditions. We have updated this paragraph to be more explicit:

- (line 235) *"Experimental values of $k_{PTR}$ are typically scarce, particularly for exact instrument operating conditions of a given set of measurements (for example, E/N of 160 Td). Instead, they can be estimated given molecular properties."*

**Line 215: I can understand how this procedure is applied to ions that are the protonated parent molecule (RH+), but how/when do we know that is true and how is this applied to a measured ion that could be a combination RH+ and fragments? (related to the question above). For example, at 69 m/Q, some fraction of this is protonated isoprene (you know this from your chromatogram) and some fraction is fragmented larger molecules. It might help the reader to walk through your procedure for an example like this on how you would extract [isoprene].**

We believe GC is truly necessary to fully understand and properly quantify the measurements. As with fragmentation (which has been discussed in previous comments), the rate constant and estimated sensitivity may not be the best aspect to modify, but rather the measured signals should be corrected where appropriate. Our response and edits in response to Referee #1's general comment address this comment as well, including an example processing procedure for isoprene from Vermeuel et al. (2023).

**Line 248: There is some strange formatting here with the inserted symbols.**

Thank you for identifying that issue, we have removed the erroneous strikethroughs.

**Line 265: What is the physical reason for transmission to decline at high mass? I would have expected this to be operating as a high (mass) pass filter?**

Our use of "transmission" in the high mass regime is not wholly correct and should include "detection efficiency". We have made this correction throughout the manuscript (lines 336, 447, 452, 664).

The fields within a quadrupole ion guide are imperfect, especially at the entrance and exit. Slower high $m/Q$ ions experience more RF cycles in these fringe fields, leading to greater losses and reduced transmission relative to lower $m/Q$ ions.

Detection efficiency requires that electrons be produced by the impact of the ion with the multichannel plate detector. The number of electrons produced depends on, among other things, the velocity of the ion (until that velocity is much greater than the threshold velocity, and it is no longer $m/Q$ dependent). We have included additional information and references with this discussion:

- (line 330) *"A high $m/Q$ mass discrimination is introduced by the quadrupole ion guides due to slower velocities and non-uniform fields near the entrance and exit of the quadrupoles ((Antony Joseph et al., 2018; Dawson, 1975; Fite, 1976; Ehlert, 1970). Additionally, aging or poor tuning of a multichannel plate detector may reduce the relative detection efficiency at higher $m/Q$, resulting in mass discrimination (Müller et al., 2014). Absolute detection efficiencies are negatively correlated with $m/Q$ when not operating the detector in saturation mode (that is, the electron cascade is in saturation regardless of the ion's $m/Q$) (Oberheide et al., 1997). Typically, PTR-TOF-MS users do not operate in saturation mode due to artefacts such as ion feedback (Pan et al., 2010). To account for reduced detection efficiency in the high $m/Q$ regime, a second, optional sigmoid function is available in the toolkit."*

**Line 380: You have used the term "spectral interference" a few times. I did not see it defined. Since there could be a few different interpretations of this, it would be helpful to clarify this at first use. My apologies if I didn't catch it.**

We have added a definition of spectral interference as it is used in this manuscript:

- (line 219) *"Here, spectral interferences refer to contributions to an analyte's quantitative ion from the fragmentation and/or adducts of other ions (for example, ethylbenzene commonly fragments to form $C_6H_7^+$, contributing additional signal to that of protonated benzene)."*

**Line 410: What is the y-intercept in the slope that is not constrained by the zero. It looks quite large. Were lower concentration calibration points done to fill in the gap between the 1-3 ppb region to assess this further?**

We have included the y-intercept:

- (line 519) *"…had a negative y-intercept of -1700±400 cps (Fig. S3; error reflects uncertainty in the linear fit)."*

We were unable to do additional calibrations at lower concentrations due to limitations in diluting our standard. We have added this detail to the methods:

- (line 161) *"This range of concentrations was limited by the possible dilution flow rates.."*

**Line 416: If diffusion is important, do the residuals scale with the diffusion constants as expected?**

We see a minor correlation between the residuals and the diffusion coefficients and included a supplementary figure:

- (line 526) *"Figure S9 shows a minor correlation between standards' average residuals and their diffusion coefficients in air (Yaws, 2008), although there are likely other factors as well."*

We do not think this is the only factor affecting the calibrations. However, we did find a change in sensitivity (~5%) when simply changing the mixing geometry of the calibration tee (cal, zero, Vocus inlet vs zero, cal, Vocus inlet). This experiment and prior experiences with mixing issues have led us to attribute part of the discrepancy to mixing. This issue was not rigorously explored, so we do not attempt to discuss it much further.

**Section 4.5: It would be helpful to include in Table S1 (or elsewhere) the average zero values for these ions. I appreciate that it could be back calculated from the LOD, but I think it would be helpful for Vocus users to be familiar with what zero (cps) can be achieved with these sources. Or perhaps add a panel to Figure 6 that has a characteristic zero spectra for the catalyst that everything is referenced to.**

We have added Table S3 which shows standards' average signals for all three clean air sources during the latter half of the field measurements (when instrument response was stable and the inlet was not becoming clogged):

- (line 625) *"Table S3 summarizes the average signals for each of these three clean air sources during the latter half of the measurement period for the standards presented in Tables S1 and S2."*

**Line 610: MeSH/DMS should show a strong diel profile due to the large difference in DMS+OH vs MeSH+OH. I'd expect if you look at the nighttime correlation it will be even stronger.**

We have attached the diurnal average of the MeSH : DMS ratio. We certainly see the effect of OH chemistry during the day. We did also investigate the nighttime (22:00 – 6:00 MDT) correlation, but found the same correlation coefficient of 0.79. We do not believe this topic requires further investigation for the present manuscript, but we thank Reviewer #1 for the suggestion for future work. We have also noted the role of chemistry in these observations:

- (line 742) "*Some of the variability in their correlation may be attributed to chemistry as methanethiol reacts ~8 times faster than DMS with hydroxyl radicals during the day (Wine et al., 1981).*"

[Figure]

**References:**

Vermeuel, M. P., Novak, G. A., Kilgour, D. B., Claflin, M. S., Lerner, B. M., Trowbridge, A. M., Thom, J., Cleary, P. A., Desai, A. R., and Bertram, T. H.: Observations of biogenic volatile organic compounds over a mixed temperate forest during the summer to autumn transition, Atmospheric Chemistry and Physics, 23, 4123–4148, https://doi.org/10.5194/acp-23-4123-2023, 2023.

---

## Author Comment (AC2)

**Response to Referee #4's Comments for: "Measurements of VOCs in ambient air by Vocus PTR-TOF-MS: calibrations, instrument background corrections, and introducing a PTR Data Toolkit"**

We thank the Editor and Referees for their time and constructive feedback on the manuscript. We have addressed all referee comments and updated our manuscript accordingly. Please find our summarized responses below. Referee comments are printed in **bold black**, our specific replies are in this blue color, quotes from the submitted manuscript are *in this blue color and italics* and quotes from the revised manuscript are provided in this *green color and italics*.

In addition to corrections made in response to Referee comments, we have made additional corrections for clarity and grammar.

We have corrected methanol's "Quantitative Ion" entry in Table S1 to reflect that the sensitivities and LODs correspond to protonated methanol summed with its water cluster (as already stated in the text, line 167).

Following discussions and troubleshooting with members of the community, we have fixed a bug in the PTR-DT which prevented it from compiling in Igor Pro 8. We have clarified which versions of Igor Pro are known to work with the PTR-DT.

- (line 200) *"Currently, version 1.1 is known to function in Igor Pro 8 and 9."*

We identified minor mistakes in Figure 6 where traces were misnamed. The scientific conclusions have not changed aside from the minor observation that *acetonitrile* experienced reduced fragmentation due to increased water clustering as opposed to *acetone* (line 600).

**Anonymous Referee #4 Responses:**

**Overview:**

**The authors have clearly spent a considerable amount of time working/learning/thinking about their GC-Vocus instrument and the interpretation of their data, and here they share important insights into how to analyse and calibrate GC-Vocus data. The authors ran a "test" (my words) campaign in Boulder, Colorado in the spring of 2021 and troubleshooted aspects of their instrument, particularly related to sensitivity and blanks, that had important impacts on their data collection. The authors wrote a tool kit for PTR sensitivity calculations that they are sharing with the community. Overall, this manuscript currently reads as a compilation of what the authors learned and how they resolved the complexities of calibrating GC-Vocus data. The authors did an excellent job referencing the literature. At the very end, the authors show some of the data they collected in Boulder with a focus on aldehydes, siloxanes and sulfur-containing compounds.**

**The work present is an important resource for the community despite this manuscript having a narrow (but growing!) audience of Vocus users, and likely more specifically GC-Vocus users. To improve this manuscript for publication, I would strongly encourage the authors to consider adopting a Standard Operating Procedure (SOP) style in order to teach the "whys" of their choices. The most likely future readers of this work will be graduate students and I expect they would greatly benefit from additional details and justifications throughout the text. I hope to have identified and described these points here as best as I can to help improve this manuscript as a resource. I anticipate this manuscript to be on the to-read list for any future/incoming (GC-)-Vocus users.**

We thank Referee #4 for taking the time to review this manuscript and for providing insightful feedback. We believe an SOP style is a good suggestion and we attempt to implement this style as we address the below comments. Additionally, suggestions from Referee #1 and clarifications for their comments aid in this effort.

In an effort to address the "whys" of some choices that may not be obvious to some readers, especially graduate students being introduced to these methods, we have included additional explanations throughout the text:

- We have moved instrument background measurement section before the calibration section to follow the order of data processing. (line 165) *"Following instrument background corrections, the fast calibrations were applied by linear interpolation…"*
- (line 158) *"The fast calibrations were buffered with 2 min. of equilibration time prior to the calibration measurement, and 2 min. of purging time with clean air from the HC trap afterwards to remove excess calibrant before recontinuing ambient measurements."*
- (line 249) *"Signals attributed to charge transfer products were not included as they represent a different ionization pathway (e.g., $O_2^+$ or $NO^+$)."*
- (line 315) *"From Eqn. (3), the y-intercept is expected to be zero. Here, the intercept is arbitrarily not forced through the origin."*

**General Comments:**

**In the spirit of making this manuscript an SOP as well, I would encourage the authors to add more details so that the presented optimized data analysis could be repeated by a future GC-Vocus user:**

- **Show all their calibrations plots where the sensitivity was calculated**
    - We are unsure what "calibrations plots" refer to in this context, but we attempt to address a few possibilities:
        i. Calibration curves – we do not see a benefit from showing all of the calibration curves (multipoint or fast) for all of our standards. One example (Fig. S3) should suffice for a simple calibration curve. We have added a reference to this figure in the methods section.
        ii. PTR-DT plots (e.g. Fig. 1) – we do not see a benefit in showing these regressions at each point in time. Rather, we show the time series of the fitting parameters (Fig. 2). In response to a later comment regarding the post-field calibrations with the PTR-DT, we have included the post-field analogue to Fig. 1 (Fig. S7).

- **Show sample time series of blanks, fast cal, long cal, GC over 2-3 cycles of their 2h procedure/ TPS script.**
    - We have added a new Fig. S2 to show 3 full cycles of the measurement cycle used here. This figure includes a panel for the full time series, as well as panels to focus on a multipoint calibration, a fast calibration, and the long background measurements.

- **Show background signals of (at least) their calibrants from which they calculated their LODs.**
    - We believe the purpose of this comment is to provide the reader with a sense of reasonable instrument background signals one might expect while operating the Vocus. To that end, this comment is partially addressed by a response to a comment by Referee #1 regarding the addition of a table of instrument background signals of our standards (Table S3). The background signals for all ions are in the database as well. Visualization of temporal behavior of these measurements are addressed between the previous and next bullet points.

- **Show zero time series. I suspect 2 mins was too short for many of the "stickier" compounds.**
    - We have included zoomed-in looks at the HC trap zeroes as part of the example time series of the 2h procedure (Fig. S2) (in relation to the comment 2 bullet points prior). We include acrylonitrile as one of our "stickier" standards. We also include a description of how instrument background signals were derived, including for "stickier" compounds.
    - (line 148) *"For species with fast responses such as acetone (Fig. S2b), the instrument background signal was derived from the median of the second half of the two-minute measurement. Acrylonitrile demonstrated somewhat longer response time (Fig. S2b). For species with longer response times, a double exponential function was fit to the data to derive the instrument background."*

- **Show the fragmentation patterns (like Fig S2) for all their calibrants.**
  - We updated the figure to include all standards (now Fig. S6).

- **Show a fitted peak of m/z 19 (How did the authors integrate m/z 19? I am skeptical of what m/z 19 can tells us in the Vocus).**
  - We did not integrate $m/z$ 19, but rather provided an example normalization against the unit mass. In normalizing, the temporal variability in $m/z$ 19 is the focus rather than absolute signal. Regardless of what "$m/z$ 19 can tell us in the Vocus", the key takeaway is that traditional normalization does not work for the Vocus – which we believe falls in line with the Referee's skepticism. We have clarified the use of UMR and added a brief explanation to the text:
  - (line 553) *"Normalization against m/Q 19 (unit mass of $H_3O^+$), m/Q 37($H_5O_2^+$, the hydronium-water cluster)…"*
  - (line 556) *"The unit mass for hydronium was used due to difficulties with accurate mass calibration and peak integration at this low m/Q."*

- **Add at table with all the dates and times and data of the full calibrations pre- during and post-campaign. (This information would be useful for others planning their field campaign calibrations timing.)**
  - The dates and times of the fast and multipoint calibrations as well as sensitivities are available in the data repository. We have added a note in the text to make this point clear:
  - (line 168) "*The measured sensitivities for both fast and multipoint calibrations are available in the data repository (see Data Availability)."*

- **Give examples of graphical linear interpolations (to help substantial/illustrate the point on lines 127-129)**
  - In the new Fig. S2, we have included a longer time series of the instrument background measurements derived from the three clean air sources. These show the linear interpolations of these measurements as applied to ambient acetone signal, as well as the differences in magnitude.

**The authors present and discuss the value "f" (the fraction of signal attributed to the quantitative ion) (line 173) as a new parameter to be considered when calculating sensitivities. But I'd like to challenge the authors on the pros and cons of this parameter as I read their manuscript:**

Referee #4 raises several important points regarding fragmentation which we address below.

- **On lines 314-316, the authors conclude that the fragmentation of the ions was constant throughout their campaign. So why go through the hassle of quantifying f then? One could just calibrate the sensitivity of the ion.**
    - If standards are available, the calculations outlined in this manuscript become unnecessary. However, we cannot simply calibrate everything we wish to quantify. Before addressing *f*, we have included this statement as the PTR-DT is introduced:
    - (line 195) *"Direct calibration of a standard is preferred, but standards are not available for all analytes one may wish to quantify. In the absence of standards for specific analytes, available standards for other analytes may be used to characterize instrument response and estimate the sensitivities of other analytes of interest."*
    - But the vast majority of the detected signals cannot be directly calibrated, making it necessary to estimate sensitivities (and thus *f*). Fragmentation can still be determined without standards via the GC for a chromatographic peak of interest. Quantifying *f* for standards is necessary to account for all ions produced in the IMR, to fully understand the ion chemistry, and to use the PTR-DT with these compounds which lack standards. We have included further discussion in Section 3.1.1.
    - (line 255) *"Values of f were determined to account for all ions produced by PTR in the IMR. While standards are directly calibrated, their fragmentation rates are necessary to characterize the simple reaction kinetics in the later stages of the PTR-DT. To estimate the sensitivities of analytes which cannot be directly calibrated, the estimated sensitivities must also be corrected for fragmentation. In the absence of direct calibration, sensitivity uncertainties resulting from ignoring fragmentation may range from negligible (acetonitrile) to a factor of 4 (decamethylcyclopentasiloxane), although fragmentation was enhanced by the high E/N in the study. In the absence of standards, if a GC is available, f can be quantified for identified chromatographic peaks. Without a GC to quantify fragmentation, an estimate from the literature or an informed assumption (e.g., from an analogous compound or from a database) may be preferred over no correction. In either case, untargeted analyses are still possible and reasonable accounting for fragmentation will improve the accuracy of quantified mixing ratios."*

- **What would be the error introduced if the authors calculated sensitivities based solely on their Vocus data (and did not have GC data available, so wouldn't be able to calculate f). This discussion is likely very relevant for Vocus users without a GC add-on.**
    - We agree that this aspect is important to make clear for the reader. We have included a brief discussion, combined with the previous bullet point. The bottom line is that Vocus

users who do not have access to the GC add-on would have to rely on the literature to determine $f$.

- **Do the authors suspect that compounds may also be fragmenting on the GC column?**
  - We do not believe thermal decomposition would impact our retrieval of $f$. We have added the relevant discussion to the text:
  - (line 282) *"Thermal decomposition during injection and chromatography is a possibility, particularly following flash heating to 300 ℃ (Fig. S4), although such products were not observed during this study. Decomposition products would be expected to arrive at different retention times than the parent and not affect the observed fragmentation rate in the Vocus itself. Additionally, losses of analyte in the GC system, related to decomposition or otherwise, would not affect f since all peak areas are relative to parent ion."*
  - Fragmentation similar to that which occurs in the IMR (formation of ions) is unlikely given the relative energies. Even so, ions would not be expected to leave the GC without further reactions.

- **Then can "f" only be calculated for known molecules that have been previously measured by PTR? What are the implications of a value such as f for untargeted analysis?**
  - This comment follows from the first two bullet points and our response is included in the first bullet point.

- **Ionization sources like EI have large databases of mass specs of pure samples where the fragmentation pattern can be used to identify unknowns. In PTR, my impression is that the parameters of the instrument vary too much from one instrument to another for such a database to be useful. Do the authors anticipate having to re evaluate their "f" factor for every campaign they run?**
  - We agree that databases for fragmentation may not have the same usefulness for PTR-MS as with EI. Pagonis et al. (2019) present a library which include fragmentation information as informed from the literature. However, fragmentation seems to vary as observed in Fig. 4, even with simply changing the PEEK capillary. We went on a field campaign in 2022 to a different location at sea level with the same instrument (same IMR settings, different tuning otherwise) and saw some similar values of $f$ and others different. We have included some discussion on this topic:
  - (line 398) *"While variable fragmentation may not always be negligible, these observations suggest that fragmentation can be treated as approximately constant under reasonably constant reactor conditions. With this assumption, fragmentation may be probed less frequently. However, for quality assurance, it would be prudent to reevaluate fragmentation rates with any significant changes such as replacing the inlet capillary, tuning the instrument, or moving the instrument. At minimum, it is*

*recommended to quantify these fragmentation rates at the beginning and end of field measurements, if possible."*

**Specific and Technical comments:**

**I encourage the authors to use additional subsections to help guide the reader to the section of interest. Each section can be constructed as a paragraph (with a topic sentence, details and summary sentence/transition sentence). For example, section 4.1 goes on for more than 3 pages without subsections, making it difficult to be used efficiently as a resource.**

We agree with this point and have broken up Sections 3.1 and 4.1–4.5. Section 3.1 was restructured (discussion of $k_{PTR}$ moved to the top) to put fragmentation in its own section.

**Title: the term GC could be included in the title since part of the novelty of this work is using the GC data.**

We have included "GC" in the title. We also included "Real-Time" to point out that both methods are used together in this work. We have updated the abstract to reflect the use of GC:

- (line 13) *"This study utilized a Vocus-PTR-TOF-MS coupled with a gas chromatograph for real-time and speciated measurements of ambient VOCs in Boulder, Colorado during spring 2021."*

**Abstract: I'm left wondering at the end of the abstract about what was observed during the field campaign in Boulder 2021, and what the authors were aiming for as research questions during this campaign?**

This is a great point; we have included brief statements in the abstract with regards to the aim of these measurements and they key results from

- (line 15) *"The aim of these measurements was to understand and characterize instrument response and temporal variability as to inform the quantification of a broader range of detected VOCs."*
- (line 33) *"Finally, applications of measurements with low detection limits are considered for a few low-signal species including sub-pptv enhancements of icosanal (and isomers; 1 min average) in a plume of cooking emissions, and sub-pptv enhancements in dimethyl disulfide in plumes containing other organosulfur compounds. Additionally, chromatograms of hexamethylcyclotrisiloxane, octamethylcyclotetrasiloxane, and decamethylcyclopentasiloxane (D3, D4, and D5 siloxanes, respectively), combined with high sensitivity, suggest that online measurements can reasonably be associated with the individual isomers."*

**Introduction: I was confused whether the authors were aiming to discuss PTR technology overall or Vocus specifically. It would be worthwhile to have a paragraph discussing how the Vocus' quadrupole on the drift tube uniquely changes the sensitivity, LODs and RH dependence for VOC detection. Such a discussion would set the stage better for the "why" this manuscript is timely. (for example line 46 should continue to discuss the different ion sources for H3O+)**

We have added a new paragraph (line 69) which focuses on the Vocus IMR (and removed the repeated information from the methods) as a transition from general PTR-MS technology, to Vocus-centric discussion, to the overview of this study.

We have also added a brief note on $H_3O^+$ ion sources:

- (line 51) *"A hollow-cathode ion source is used in most PTR-MS instruments, but Tofwerk's Vocus uses a conical, low-pressure discharge source (Krechmer et al., 2018)"*

**Lines 41-42: I would argue that the logic is reversed here, and that it's rather the technological advances that lead to new VOCs detected and new LODs/sensitivities achieved.**

We agree with this sentiment and we believe that it was the original intention of our (poorly framed) statement. We have re-worked this statement:

- (line 45) *"As the atmospheric chemistry community continues to investigate VOC emissions and their chemical evolution, technological advancements have allowed the detection and quantification of a broader range of VOCs at lower concentrations."*

**Line 47: change "functional group" to heteroatom, since a methyl group is considered a functional group on a molecule but wouldn't have a proton affinity higher than that of water.**

We have made this change (line 54).

**Line 48: I'm not sure I followed why fragmenting alkanes are "most notably" in the context of PTR.**

Alkanes would be the "most notable" class of compounds that are not well-measured by PTR-MS due to their abundance. We have changed this sentence to be more explicit in meaning:

- (line 54) *"PTR-MS is insensitive to alkanes due to inefficient proton transfer followed by fragmentation (Gueneron et al., 2015)."*

**Paragraph from lines 69-76 is subjectively redundant, and I would delete it to help make the writing more concise.**

We agree with this suggestion. Rather than remove the paragraph entirely, we combined the key details with the prior paragraph:

- (line 80) *"This study details the quantification of VOCs measured in Boulder, Colorado in spring 2021 with a Vocus-2R PTR-TOF-MS, hereafter referred to as the Vocus, and presents an open-source PTR Data Toolkit (PTR-DT). This study aims to address the production of reliable ambient measurements made by PTR-MS, particularly regarding species lacking standards for calibration. The PTR-DT was used to derive instrument characteristics from measured standards and to estimate the sensitivities of additional species on the timescale of frequent calibrations (every 2 h in this study). Additionally, these frequent calibrations were explored as an alternative for the normalization of ambient measurements against the reagent ion signal, including a time period of changing ion chemistry in the IMR. Three sources of clean air for instrument background measurements were compared to identify which source yielded the lowest limits of detection. Finally, some findings are presented to demonstrate the low detection limits."*

**In their operating conditions, why did the authors run their IMR at 90 degs and a slightly higher pressure of 1.5 mbar (compared to 1.2 mbar). What would these parameters help optimize?**

We have included a brief explanation for these operating conditions:

- (line 97) *"Warmer IMR temperatures improve delay times of species with lower volatilities (Mikoviny et al., 2010). The IMR axial voltage and pressure, and thus E/N, were chosen to limit the formation of water clusters and promote simple reaction kinetics, at the expense of analyte fragmentation."*

**The authors share that their instrument had to be troubleshooted during their field campaign. I would suggest that the authors add a subsection called troubleshooting in their methods and provide more details on their ion source malfunction (for example on line 102) and their capillary clogging (and how did they unclog it?)**

We have moved these two paragraphs regarding troubleshooting to a new Section 2.2 (Instrument troubleshooting). We have included more details regarding the ion source malfunction:

- (line 120) *"The ion source malfunctioned and was unstable from April 9–10 and 20–21,. The electrical current supplying the ion source was highly variable on second timescales and demonstrated a step-change toward higher currents. We believe this was indicative of an incomplete ring in the conical discharge ion source. This issue was resolved by turning off the ion source and flow of water vapor for several minutes before returning to normal operation. The cause is unknown, but a water droplet entering the ion source may be one possibility. Since this discharge was responsible for forming primary hydronium ions, the consequences of this malfunction were different ion chemistry and reduced sensitivities. changing the ion chemistry in*

> *the reactor and reducing sensitivities. For demonstrative purposes, these time periods were*
> *included in the discussion of the PTR-DT but are excluded in the final quantified dataset."*

- (line 371) *"After disassembling inlet to replace the capillary and resetting the ion source*
  *voltages, the ion chemistry... "*

We did not unclog our capillary, but rather removed and replaced it entirely. We had previous issues where cleaning the capillary with methanol (one old recommendation) would only be a temporary fix.

- (line 374) *"Prior solutions have involve using a solvent to clean the capillary or blowing it out*
  *with nitrogen, but, at the time of writing, Tofwerk AG recommends replacing the capillary. The*
  *capillary was removed and replaced on April 10$^{th}$ ..."*

**On the section of the clogged capillary: why would a change in flow impact the sensitivity? The mixing ratio does not change and the flow of water remains small compared to the inlet flow. In other words, what do the authors see as the implications of their discussion on lines 105-110?**

The flow of water vapor is not small for the Vocus. We have not included any discussion for the impact of the clogged capillary on sensitivities here. Instead, we have noted the implication as to why this changing flow rate matters and point to the relevant section:

- (line 133) *"The obstruction and changing flow rates were coincident with changing ion chemistry*
  *due to the reagent water vapor flow (15 sccm) and the transition from 7% to 23% water vapor by*
  *volume in the IMR, as discussed in Section 4.4. To avoid obstructions of the inlet capillary, we*
  *recommend adjusting the geometry of the inlet such that the sample and bypass lines are*
  *perpendicular to the Vocus inlet (Fig. S1b)."*

We have also include the water flow rate in the methods section:

- (line 95) *"The ion source was supplied with a 15 sccm flow of water vapor."*

**Why would methanol (lines 119-120) be the only VOC here to have a water-cluster relevant for its quantification? I'm not sure I followed this argument, or the uniqueness of methanol in this case.**

We used the water cluster to improve sensitivities since protonated methanol was strongly attenuated by the BSQ while the water cluster, having a higher $m/Q$, had a better transmission efficiency. The signal of the water cluster was greater than that of the protonated product, which is unique to methanol for the standards used in this study. We have attempted to clarify the text:

- (line 166) *"Methanol was strongly attenuated by the BSQ, resulting in very low sensitivity. To*
  *enhance its sensitivity, the signals of protonated methanol its cluster with water were summed."*

**Thuner is an important part of Vocus data analysis and it isn't mentioned anywhere in the text. I would encourage the authors to discuss this procedure in detail in their sensitivity discussion.**

We have included a brief discussion of tuning in Section 2.1:

- (line 104) *"Prior to all measurements, the instrument's signals were optimized. With a constant flow of a standard mixture, voltages for the ion optics between the IMR and the TOF mass analyzer were coarsely adjusted to improve overall signal. Finer adjustments were made via Tofwerk AG's Thuner software (v1.13.0.0) which programmatically adjusted voltages and analyzed relative sensitivities and mass resolution. A setpoint was chosen which compromised between high mass resolution and sensitivities."*

**Methods, section 3: how do the authors calculate/determine the initial concentration of H3O+? Isn't the Vocus blind to the reagent ion?**

We do not determine the initial concentration of hydronium for this study as it is not necessary. Instead, $[H_3O^+]$ and the reaction time, $t$, are part of the slope in Step C of the toolkit. Following Eqn. (3), the toolkit essentially plots sensitivities which have been corrected for fragmentation (and only species with $T(m/Q) \sim 1$ are used) against $k_{PTR}$:

$$\frac{S_{inst}}{f \times T(m/Q)} \approx ([H_3O^+]_0 \times t) \times k_{PTR}$$

We have clarified this point in Section 3.3:

- (line 311) *"The slope of the resulting linear fit is the product $[H_3O^+]_0 \times t$ such that neither quantity is necessary to complete these calculations."*

Separately, the Vocus isn't completely "blind" to the reagent ion. The transmission of hydronium is significantly reduced, but there is sufficient signal (~1000s cps) to observe trends. If we were to use it for instrument normalization, however, it would not correct for the observed trends in measured sensitivities and instead introduced additional, undesired trends (Fig. S8).

**Methods, section 3: equations 1, 2 and 3 do not have consistent definitions. Would "S" be missing in Eq 1? S and Sinst cannot both be equal to [RH+]/[R]. It might be worth cleaning up this section, and providing a solution to these questions for one compound of choice (a compound discussed in section 4.6 for example?)**

Eqn. (1) is correct (not missing S) and appears throughout the literature (Hansel et al., 1995; Hayward et al., 2002; de Gouw et al., 2003; Yuan et al., 2017). We do not believe it is inconsistent with Eqns. (2) and (3). Sensitivity, the number of ions produced per amount of analyte injected, in Eqn. (2) is an outside parameter defined in terms of Eqn. (1).

To rectify Eqns. (2) and (3), we have updated Eqn. (3) to use $[RH^+]_{meas}$, the ions measured, in place of $[RH^+]$, the ions produced in the IMR.

**Lines 179-184 – could there be a picture of the interface/code that could be included here to support visually the contribution of the authors to developing the PTR-DT?**

We have included screenshots of the interface for steps B, C, and D (Fig. S5).

- (line 224) *"Screenshots of the interfaces for Steps B–D are shown in Fig. S5."*

**Line 319: why did the sensitivity increase?**

We have included the brief explanation and pointed the reader toward the more detailed discussion later in the text:

- (line 406) *"The slope then increased due to generally increasing sensitivities in the field, driven by changing ion chemistry as discussed in Section 4.4, then …"*

**Line 329 and 337: why did benzene and toluene have a different response to changing ion chemistry?**

Different species responded to the changing ion chemistries in different ways, as is described further in Section 4.4. The water cluster distribution, for example, will impact different standards according to their proton affinities as is described in this paragraph. We have attempted to clarify this point here:

- (line 417) *"That is, benzene and toluene had a different response to the changing ion chemistry relative to acetone and other standards, likely due to, in part, the relative reaction rates with hydronium and water clusters which are changing during this time. The changing ion chemistry is discussed in greater detail in Section 4.4."*

**Line 366-367: specify which standards were included and which were omitted.**

We have added this information:

- (line 459) *"All standards in Tables S1 and S2 with fragmentation information and no known spectral interference were included (excluded: methanol, ethanol, phenol, t-amyl ethyl ether, β-caryophyllene)."*

This comment also helped us to identify an error which was left over from a previous version of the manuscript. We had noted that species with no spectral interferences were included in this histogram (line 366 in the original text), but some species with interferences were included. We have not changed the figure as the inclusion of these species was intentional. Instead, we call direct attention to their inclusion:

- (line 461) *"However, this histogram includes species with known spectral interference (toluene, acetone in the same standard mixture as t-amyl ethyl ether, and both benzenes in Table S2) which have high residuals."*

**My understand of the Vocus technology (Krechmer et al. 2018) is that there is no RH dependence due to much larger concentration of H3O+ ions in the drift tube. I was therefore confused by the discussion on lines 371-373, since the authors go on lines 418 to specify there was no dependence on saturation vapour concentration (or do they mean ambient H2O?).**

Referee #4 is correct that there is no RH dependence for PTR due to the large abundance of hydronium. Formaldehyde (and others), however, also undergoes the reverse reaction. Previous studies have shown a humidity dependence for this reverse reaction (Vlasenko et al., 2010). This dependence in the Vocus with its high water mixing ratio, to our knowledge, has not been investigated. We suspect a strong reverse reaction due to the abundance of water vapor, but perhaps negligible dependence on relative humidity. We have removed "humidity dependence" from this line and explicitly noted this knowledge gap:

- (line 468) *"This back reaction requires further investigations with regards to the Vocus due to the abundance of water vapor in the IMR."*

We believe there was a misunderstanding with "saturation vapor concentrations" as this was in reference to the standards rather than water vapor. The lower volatility compounds (with lower saturation vapor concentrations) may be expected to interact with the surfaces during calibrations, leading to lower measured signals and larger (more negative) residuals during the fast calibrations. We have attempted to clarify this statement:

- (line 528) *"Lower volatility standards demonstrated greater residuals, but no clear dependence on standards' saturation vapor concentration was observed and inlet passivation does not seem to be the primary cause for the residuals."*

**Section 4.3. based on this discussion, can the authors speak to the value of introducing an internal standard into the Vocus (which wouldn't be present in ambient air).**

The use of internal standards is a great suggestion (and one we have discussed with other colleagues). We have added a new paragraph:

- (line 563) *"One possible modification to this method of fast calibrations may involve a constant introduction of an internal standard which is otherwise not present in the sampled air, e.g. deuterated acetone or benzene. This method, in a sense, brings fast calibrations to the extreme of "calibrating" with each mass spectrum. Further investigation and validation is required. While this method was not used in this study, we make a few suggestions following our observations. Multiple standards may be necessary to account for different dependencies on hydronium and its water clusters. Additionally, care should be taken in assessing fragmentation of such standards and how they may interfere with analytes. Finally, we strongly encourage the use of internal standards in addition to traditional calibrations with more standards."*

**I worry that the acetone artifacts discussed on lines 454-455 are related to acetone present internally inside in the instrument, and not due to the calibration...**

We observed similar trends for most signals as we saw for acetone and benzene (which served as examples of everything in between). We would expect an elevated signal within the instrument to manifest as a higher instrument background rather than a change in sensitivity (it would become something similar to a standard addition). We have modified the text to make it clear that these observations were not unique to acetone and benzene:

- (line 578) *"Most other standards and ambient signals saw similar trends to varying degrees of severity."*

**Lines 479-484: I didn't quite follow this discussion on ion mobility. Do the authors mean that as the flow reduced, the $H_3O^+$ flow made of a larger portion of the total flow?**

At a lower sample flow, the gas mixture in the IMR contains more *water vapor*. Since water is polar it has an outsized effect on the mobility in the mixture as explored in the discussion. It is also true that hydronium did make up a larger portion of the total flow, but we would expect all sensitivities to scale similarly with increased primary ion. We have attempted to add clarifying information in the text:

- (line 605) "*The mixing ratio of water vapor in the IMR increased three-fold, increasing the average polarity of the buffer gas and increasing the frequency of hydronium's ion-dipole interactions.*"

**Figure 1b: Could the authors clarify how/why many compounds were not included in the fit and how the authors went about deciding which compounds were excluded from the fit. This information would be important to avoid data manipulation.**

The rationale for excluding standards was described in Section 3 as each subplot (and step of the PTR-DT) was addressed. We have added text to point the reader to Section 3 for more information:

- (line 1080) *"Grayed-out standards were excluded from the respective fits as described in Section 3."*

**Figure 1a: what did the post-field campaign comparison look like?**

The post-field comparison (to the first field sensitivity measurement) was very similar. We have included a new figure in the SI (essentially a copy of Fig. 1, except comparing against the post-field calibrations) and included references to it in the main text:

- (line 294) *"... (Fig. S7a shows a similar regression using post-field laboratory calibrations). Fits using pre- and post-field calibrations yielded similar results. The post-field calibrations had lower sensitivities, yielding a higher slope"*
- (line 313) *"Figures 1b and S7b shows an orthogonal distance regressions for the first field calibration using field-estimated sensitivities of laboratory standards informed from the pre-field and post-field laboratory calibrations, respectively, each yielding similar results."*
- (line 339) *"Figures 1c and S7c shows an example transmission functions derived from the first field calibration using field-estimated sensitivities of laboratory standards informed from the pre-field and post-field laboratory calibrations, respectively, each yielding similar results."*
- (line 1080) *"Figure S7 shows similar fits using the post-field laboratory calibrations."*

**Figure 2: which compounds is this data representing?**

Figure 2 shows the fitting parameters in Fig. 1 as a function of time (Fig. 1 shows one point in time), so these plots are derived from all standards (minus the exceptions outlined in the text). We have added this detail to the figure caption:

- (line 1083) *"Time series of PTR-DT fitting parameters derived using the standards in Tables S1 and S2 with the exceptions outlined in Section 3."*

**Figure 3: the use of measured/derived values is confusing and should be rewritten with clarity.**

We have attempted to clarify this figure caption:

- (line 1087) *"Relative residual histograms (5% bins) with fits to a normal distribution (average, x0 and standard deviation, σ, provided with fitting uncertainties) comparing standards' measured field and laboratory sensitivities (B), measured (or field-estimated in the case of laboratory standards) and fit sensitivities (C), measured and fit transmission (D), as well as measured sensitivities and those calculated using the parameters and regressions from the PTR-DT (E) for each fast calibration."*

**Figure 6: what are the red traces?**

The red traces indicate a ratio of <1, meaning the UHP nitrogen or HC trap outperformed the catalyst. We have clarified this in the figure caption.

- (line 1111) *"For ratios where the catalyst performed better, the trace is black. For ratios where the UHP nitrogen (a) or HC trap (b) performed better, the trace is red."*

**According to Figure 7, the LODs of the standards would appear to be biases high compared to the 616 species quantified. Why might that be?**

The vast majority of these compounds are *semi*-quantified due to unconstrained fragmentation and overestimated sensitivities. This explanation was somewhat buried in the text:

- (line 532) "*However, many of these species were only semi-quantified due to unconstrained transmission in the high m/Q regime (>300 Th) and undetermined fragmentation, leading to underestimated LODs.*"

We have included similar information in the figure caption to aid the reader:

- (line 1116) "*LODs at high mass are biased low due to unconstrained fragmentation while the siloxane standards provide a more realistic LOD limit.*"

**References:**

de Gouw, J. A., Goldan, P. D., Warneke, C., Kuster, W. C., Roberts, J. M., Marchewka, M., Bertman, S. B., Pszenny, A. a. P., and Keene, W. C.: Validation of proton transfer reaction-mass spectrometry (PTR-MS) measurements of gas-phase organic compounds in the atmosphere during the New England Air Quality Study (NEAQS) in 2002, Journal of Geophysical Research: Atmospheres, 108, https://doi.org/10.1029/2003JD003863, 2003.

Hansel, A., Jordan, A., Holzinger, R., Prazeller, P., Vogel, W., and Lindinger, W.: Proton transfer reaction mass spectrometry: on-line trace gas analysis at the ppb level, International Journal of Mass Spectrometry and Ion Processes, 149–150, 609–619, https://doi.org/10.1016/0168-1176(95)04294-U, 1995.

Hayward, S., Hewitt, C. N., Sartin, J. H., and Owen, S. M.: Performance Characteristics and Applications of a Proton Transfer Reaction-Mass Spectrometer for Measuring Volatile Organic Compounds in Ambient Air, Environ. Sci. Technol., 36, 1554–1560, https://doi.org/10.1021/es0102181, 2002.

Pagonis, D., Sekimoto, K., and de Gouw, J.: A Library of Proton-Transfer Reactions of H3O+ Ions Used for Trace Gas Detection, J. Am. Soc. Mass Spectrom., 30, 1330–1335, https://doi.org/10.1007/s13361-019-02209-3, 2019.

Vlasenko, A., Macdonald, A. M., Sjostedt, S. J., and Abbatt, J. P. D.: Formaldehyde measurements by Proton transfer reaction – Mass Spectrometry (PTR-MS): correction for humidity effects, Atmospheric Measurement Techniques, 3, 1055–1062, https://doi.org/10.5194/amt-3-1055-2010, 2010.

Yuan, B., Koss, A. R., Warneke, C., Coggon, M., Sekimoto, K., and de Gouw, J. A.: Proton-Transfer-Reaction Mass Spectrometry: Applications in Atmospheric Sciences, Chem. Rev., 117, 13187–13229, https://doi.org/10.1021/acs.chemrev.7b00325, 2017.